# High-resolution gridded estimates of population sociodemographics from the 2020 census in California

Nicholas J. Depsky[1]*, Lara Cushing[2], Rachel Morello-Frosch[3]

1 Energy and Resources Group, University of California, Berkeley, Berkeley, California, United States of America, 2 Department of Environmental Health Sciences, Fielding School of Public Health, University of California Los Angeles, Los Angeles, California, United States of America, 3 Department of Environmental Science, Policy and Management and School of Public Health, University of California, Berkeley, Berkeley, California, United States of America

* njdepsky@berkeley.edu

**Data Availability Statement:** Data are available from: https://zenodo.org/badge/latestdoi/434382697, DOI: 10.5281/zenodo.5874927, github.com/njdepsky/CA-POP.

## Abstract

This paper introduces a series of high resolution (100-meter) population grids for eight different sociodemographic variables across the state of California using data from the 2020 census. These layers constitute the 'CA-POP' dataset, and were produced using dasymetric mapping methods to downscale census block populations using fine-scale residential tax parcel boundaries and Microsoft's remotely-sensed building footprint layer as ancillary datasets. In comparison to a number of existing gridded population products, CA-POP shows good concordance and offers a number of benefits, including more recent data vintage, higher resolution, more accurate building footprint data, and in some cases more sophisticated but parsimonious and transparent dasymetric mapping methodologies. A general accuracy assessment of the CA-POP dasymetric mapping methodology was conducted by producing a population grid that was constrained by population observations within block groups instead of blocks, enabling a comparison of this grid's population apportionment to block-level census values, yielding a median absolute relative error of approximately 30% for block group-to-block apportionment. However, the final CA-POP grids are constrained by higher-resolution census block-level observations, likely making them even more accurate than these block group-constrained grids over a given region, but for which error assessments of population disaggregation is not possible due to the absence of observational data at the sub-block scale. The CA-POP grids are freely available as GeoTIFF rasters online at github.com/njdepsky/CA-POP, for total population, Hispanic/Latinx population of any race, and non-Hispanic populations for the following groups: American Indian/Alaska Native, Asian, Black/African-American, Native Hawaiian and other Pacific Islander, White, other race or multiracial (two or more races) and residents under 18 years old (i.e. minors).

## Introduction

Understanding the spatial distribution of human populations is integral to civic and land use planning, public policy design and various fields of academic research. For example, many

**Funding:** This study was funded by the California Air Resources Board (# 18RD018- RM-F and NJD), the Strategic Growth Council (CCRP0022 - RM-F, NJD and LC) and U.S. Environmental Protection Agency (#84003901 LC, RM-F and ND). The funders had no role in study design, data collection and analysis, decision to publish, or preparation of the manuscript.

**Competing interests:** The authors have declared that no competing interests exist.

public health studies in the United States (U.S.) seek to quantify the number of people residing near a potential environmental health hazard [1–4]. Similarly, environmental justice and equity oriented research often evaluates the degree to which people of color and other socially disadvantaged populations live in closer proximity to environmental contaminants or hazards [3, 5, 6]. However, the ability to estimate fine-scale spatial distributions of populations in many prior studies has been limited to the spatial granularity of population estimates that are made available by public enumerating agencies, such as the U.S. Census Bureau.

In the U.S., the most granular spatial units of enumeration are census blocks, available in each decennial year (i.e. 2000, 2010, 2020). In non-decennial years, the finest scale estimates are made at the block group level, which are coarser than census blocks. In California (CA), for example, blocks have an average land area of roughly 0.8 km$^2$ (~200 acres) and a population of about 75 people each, while block groups are roughly twenty times larger, both in average area and population. Census blocks therefore provide population information at a high spatial resolution, although in more sparsely populated regions, their areal extents tend to be much larger, consisting of large open, unpopulated spaces. Without more precise information about the likely locations of population within these areas, researchers are often forced to assume that populations are uniformly distributed across the entire area of the given census spatial unit [5, 7, 8]. Such simplifying assumptions may have significant implications on study findings, especially for research in rural areas and concerned with precisely quantifying populations within an area smaller than local census block or block group areas, such as a specified buffer distance surrounding a polluting facility [5, 9, 10].

To address this issue, many techniques have been developed to disaggregate population estimates to finer scales. Broadly speaking, this field of population downscaling is a form of 'dasymetric' mapping, a methodology which dates back many decades [11–13]. Eicher and Brewer (2001) [14] formalized many of the techniques and terminology used in modern dasymetric mapping studies in their study to disaggregate population from the 1990 U.S. census from 159 different counties. They refer to the county boundaries where they have observed population estimates as their "source zones", then mask out areas likely to be unpopulated within each county based on higher resolution, ancillary land use datasets, with final populated area boundaries within each county deemed their "target zones". Many subsequent studies emulated the dasymetric mapping techniques detailed in this study, usually employing various land use datasets as their primary source of ancillary data to reapportion population within source zones (e.g., [15–17]. Some studies construct multi-class weighting schemes to reapportion population to target zones based on the characteristics of the land use type (e.g. high-density versus low-density residential) [18–20], and often integrate additional ancillary datasets, such as tax parcel data [21, 22], home address [17, 23], property records [24], building footprints [25] and/or mobile phone data [26].

More recently, researchers have begun to employ more complex machine learning techniques to predict fine-scale population distributions within source zones, often using a wide array of ancillary datasets, such as road networks, nighttime lights, infrastructure and building footprint data, in addition to land use layers as covariates in the models [27–32]. These highly-modeled approaches can represent a significant improvement from simpler techniques, especially in regions of the world for which source zone population estimates from official census surveys are infrequent and/or only exist at very coarse spatial resolutions [33–37]. Leyk et al. (2019) [11] provide a thorough review of dasymetric mapping methods employed in past studies, including these highly-modeled approaches, to construct large-scale (i.e. global, continental) grids of population.

In this paper, we introduce a new suite of publicly available population grids, known as 'CA-POP', for the State of California produced using dasymetric mapping methods. The grids

represent values for eight different demographic variables from the 2020 U.S. Census and are provided at a pixel resolution of 100 meters. Census blocks from the 2020 census were utilized as source zones, with high-resolution residential tax parcel boundaries and remotely-sensed, individual building footprints used as ancillary datasets to construct target zones of population within each block. A relatively simple, polygon binary method [14] of reapportioning population from the block level source zones to parcel and/or building level target zones was utilized. A qualitative comparison to a few of the more heavily-modeled gridded products available in California (e.g. LandScan, WorldPop) revealed that CA-POP performs very well in differentiating between populated and unpopulated regions, comparatively. This suggests that in contexts where both source zone population estimates and ancillary datasets are available at high resolutions, simpler, more easily-replicable dasymetric mapping techniques can yield high quality grids without needing to employ more complex algorithms.

Producing high-resolution CA-POP grids for various demographic variables estimated in the 2020 census, including racial and ethnic subgroups, can serve as a resource for studies that seek to evaluate these communities. A precursor to the 2020 CA-POP grids developed by the authors, based on 2017 American Community Survey block group and 2010 block source zones, were employed by Casey et al. (2021) [38] to assess social inequalities in residential proximity to large methane-emitting sites and in Pace et al. (2022) [39] to estimate racial/ethnic inequalities in estimated drinking water concentrations of arsenic, nitrate, and hexavalent chromium from community water systems and areas of potentially high domestic well prevalence, demonstrating the utility of CA-POP for environmental equity studies. As more sociodemographic variables are released by future U.S. Census Bureau's American Community Surveys from the U.S. Census Bureau, additional grids based on these block group-level values may be produced and uploaded to the public CA-POP repository (github.com/njdepsky/CA-POP).

## Data and methods

In conjunction with the estimates of census block populations from the 2020 U.S. Census, two sources of ancillary data were used that represent spatial units at a sub-block level of spatial granularity: i) residential tax parcel boundaries and ii) estimates of the individual footprint of every building throughout the state. Both of these ancillary data sources were used to identify areas within each census block likely to contain populated, residential areas, as opposed to vacant, commercial or other non-residential space.

### Census data

We utilized block level estimates of population collected during the 2020 U.S. Census–the highest spatial-resolution available from the U.S. Census Bureau–from the (P.L. 94–171) Redistricting Data Summary File [40]. The tabular block-population data for the Summary File, as well as the shapefile of block boundaries were obtained from the U.S. Census Bureau in November 2021 for the entire state of California [41]. Specifically, data were obtained for the following variables: i) total population, ii) Hispanic/Latinx population of any race, iii) non-Hispanic/Latinx populations for all major racial subgroups available in the P.L. 94–171 file and, iv) population of minors (younger than 18 years old) (Table 1). We produced grids for all racial/ethnic subgroups made available thus far for uniform (single) race classifications, with respondents identifying as another race or as multiple races grouped into a grid for "other/multiracial" residents.

In choosing our racial/ethnic groupings, we sought to maximize the utility of CA-POP for research employing race as a proxy for experiences of racism—particularly racism operating at

**Table 1. Census variables represented as CA-POP grids.**

| 2020 Census Block Level Population Totals Obtained (P.L. 94–171 Code) | CA-POP Grid Name |
|---|---|
| *Grids created for each variable*: | |
| Population (*P002001*) | TOTAL |
| Hispanic or Latinx (*P002002*) | HISP |
| Non-Hispanic or Latinx, White (*P002005*) | NHWHITE |
| Non-Hispanic or Latinx, Black or African American (*P002006*) | NHBLACK |
| Non-Hispanic or Latinx, American Indian and Alaska Native (*P002007*) | NHAMIND |
| Non-Hispanic or Latino, Asian (*P002008*) | NHASIAN |
| Non-Hispanic or Latinx, Native Hawaiian and Other Pacific Islander (*P002009*) | NHHIPI |
| *OTHER/MULTI grid is the combined sum of*: | NHOTHERMULTI |
| Non-Hispanic or Latinx, Some Other Race alone (*P002010*) + Non-Hispanic or Latinx, Population of two or more races (*P002011*) | |
| *MINORS grid (population < 18 years old) created from*: | MINORS |
| Population of adults (*P003001*) (subtracted from P002001) | |

institutional and structural levels—to determine opportunities and risk factors at the neighborhood level. This is in keeping with the understanding of race as a social construct that has been used to systematically discriminate against and socioeconomically marginalize specific groups of people [42, 43]. We chose groupings that are typical in the environmental justice literature and somewhat reflect shared forms of discrimination [44]. However, we recognize that forms of discrimination vary widely between racial and ethnic groups that we have grouped together (for example, different immigration policies for Mexicans and Cubans, who might both identify as "Hispanic" or "Latino/Latinx"). We were limited in our ability to create more fine-grained categories due to the availability of current data, and grids for additional racial categorizations provided in subsequent 2020 Census or American Community Survey tables (e.g., additional sub-categories for Hispanic/Latinx and Asian respondents) can be generated when these data are released.

The official population enumerated in the 2020 Census for the entire state of California is 39,538,223 people across 519,723 census blocks, with a mean area of 0.79 km$^2$, or 195 acres. Estimates for each of the above values at the block-group level for the 2020 census were also obtained for use in an accuracy assessment of the dasymetric mapping method employed for total population. Block-groups are at a significantly coarser spatial resolution than blocks (~1:20), with a total count of 25,607 and mean area of 16.0 km2, or 3950 acres.

## Residential parcel data

We utilized boundaries for all tax parcels in California from LightBox-Digital Map Products (accessible at digmap.com/platform/smartparcels/), which contains 12,728,980 parcels classified by 278 different land use types. This dataset is used by the California Air Resources Board, among other state agencies, and updated quarterly. The data we utilized for this study was from the final quarter of 2018 and represents tax parcels that were assessed either in 2018 (55% of total) or 2017 (45%). Although the vintage (i.e. date of data collection) of these parcel boundaries is not perfectly consistent with the 2020 census population estimates, obtaining ancillary data with uniform vintages is challenging and rarely done in a completely harmonized manner [15, 30, 35]. Given the relative recency of this parcel data, it is still a valuable source of ancillary data for dasymetric mapping of 2020 census populations.

We identified 30 of these 278 land use classes as residential for use as ancillary data in creating the population grids; 8,839,658 residential parcels represented roughly two-thirds of all

parcels statewide, and covered 6.7% of the total area represented in the full parcel dataset. The full list of these residential land use classes is shown in the S1 Table. The average residential parcel area is 3,500 m$^2$ (~38,000 ft$^2$, ~0.86 acres), approximately 220x smaller than the average census block, making these parcel boundaries valuable for downscaling population estimates within blocks. The highest proportion of residential parcel types are 'SINGLE FAMILY RESIDENTIAL' (n = 7,255,233, 82.1%) and 'CONDOMINIUM (RESIDENTIAL)' (n = 330,047, 3.73%).

In terms of area, the most abundant land use types are 'SINGLE FAMILY RESIDENTIAL' (12,850 km$^2$, 41.5%) and 'RURAL RESIDENCE (AGRICULTURAL)' (12,000 km$^2$, 38.8%). The amount of populated residential area within each parcel varies greatly, especially between certain land use types, such as 'SINGLE FAMILY RESIDENTIAL' and 'RURAL RESIDENCE (AGRICULTURAL)'. For example, the former tends to be fairly small, encompassing a single house and surrounding lot area, while the latter often includes a farm residence as well as adjacent agricultural fields. Therefore, even within many residential parcel boundaries, there is a need to further distinguish populated versus unpopulated space, which we largely achieve here through the use of building footprint data.

## Building footprint data

Further distinguishing between open space and populated areas within blocks and larger residential parcels was done using publicly-available, remotely-sensed building footprints produced by Microsoft for the entire country. The initial version of this dataset was released in 2018, though a second version was released in early 2021 and was obtained in November of 2021 for use in this study. These building footprints were identified from publicly-available satellite imagery of the U.S. and employed a series of machine learning (deep neural net) classification algorithms to identify likely building rooftops, converting these footprints to a polygon shapefile for each state. More information on the production of this dataset can be found on its online source repository (github.com/microsoft/USBuildingFootprints). This dataset contains estimated footprints of 11,542,912 distinct buildings across California, with an average individual building area of 277 m$^2$ (~2980 ft$^2$, ~0.07 acres), or approximately 13x smaller than the average size of residential parcels, and ~2,850x smaller than the average census block area statewide. The approximate date range of the source satellite image used to create each building footprint is also provided, with 91.6% of all buildings delineated using imagery from 2018 or later.

Despite being the ancillary data source of highest spatial granularity, one inherent limitation of the building footprint data is that it is a single-class dataset, with no distinction between building types, making it difficult to identify which buildings are residential structures. Additionally, the classification algorithm used for building delineation is not perfect, with Microsoft reporting its accuracy in terms of precision and recall at 98.5% and 92.4%, respectively. Precision pertains to relative error rates of false positives (detecting a building where there is none), suggesting a false positive rate of 1.5%, while recall pertains to false negative error rates (failing to detect an existing building), suggesting a false negative rate of 7.6%. This rate of false negatives is not insignificant, and examples of such instances can be seen (S1 Fig). Given the limitations of the building footprint data, we opted to use entire parcel boundaries to represent populated areas in small residential plots to avoid completely relying on building footprints to identify populated structures within all residential parcels, described in further detail below. However, the relative performance of Microsoft's building detection algorithm is still remarkable and their footprint dataset allows for substantial spatial downscaling of likely residential zones, especially in areas where fine-scale residential parcels are absent. Fig 1 presents examples of census block group boundaries and the ancillary datasets.

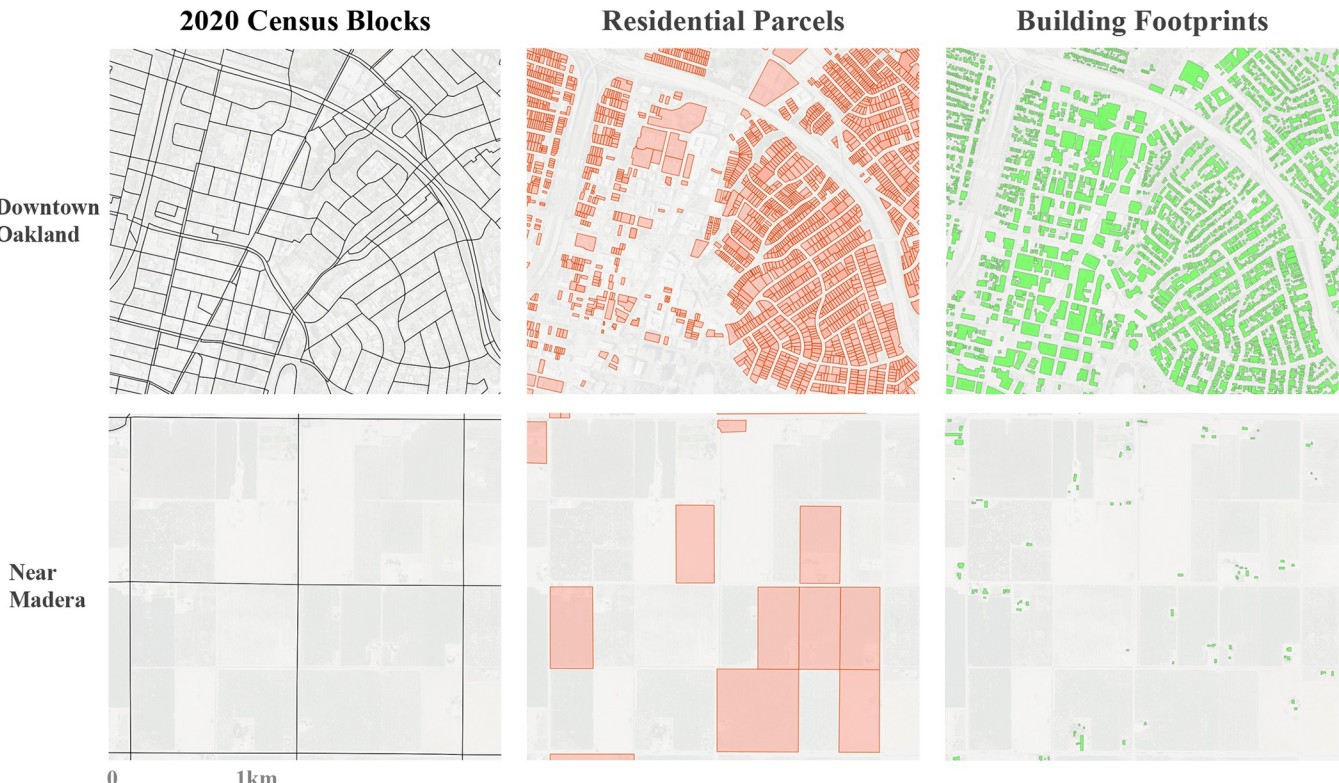

**Fig 1. Data sources used for the population grid creation process.** Examples are shown in in urban (top row) and rural (bottom row) settings. The 2020 census blocks represent the source zones of population and the parcel and building footprint data the ancillary data comprising the target population zones. (Satellite base-imagery source: USGS (NAIP) from The National Map).

### Dasymetric methods

Using the 2020 census blocks as source zones for population estimates across California, the residential parcel and individual building polygons were used to apportion population to smaller sub-regions within each block. Therefore, some combination of residential parcels and/or building boundaries served as the target zones for population within each census block, with this final vector layer of populated areas then converted to a 100m-resolution statewide grid. This approach could be classified as a form of 'polygon binary' dasymetric mapping, where vector polygon ancillary data sources are used to define populated versus unpopulated classifications within source zones, assuming population is homogeneously distributed amongst the populated target zones regions within each source zone [14].

Other, more complex dasymetric mapping techniques that utilize multi-class information associated with ancillary data to assign population density weights to each land use region have been employed in past studies as well [15, 18, 22]. In theory, a similar approach could have been employed with the residential parcel ancillary data used in this study, assigning relative population density weights for each of the 30 residential parcel types. However, coming up with appropriate weights for each parcel type is not straightforward, especially as many of the residential parcel classes are absent or rare in some counties compared to others. Also, many of the studies that apply multi-class weighting schemes utilize population source zones and ancillary land use datasets at much coarser resolutions than the data sources we used (e.g. census tracts rather than blocks) [14, 22, 25]. Accounting for likely variation in population densities between parcel types is less important with smaller source zones like those used in our

study. Furthermore, given that the building footprint data lack classifications of structure type and the fact that both parcel and building boundaries were often both utilized to apportion population within a given census block, we opted to treat both ancillary data sources in a binary fashion.

Creation of the 100m x 100m statewide grids from the 2020 census population estimates was done for each census block, in a stepwise fashion as follows:

A) *Identified all "small" residential parcels to include as eventual target zones in each census block.* "Small" residential parcels were defined as those with an area less than or equal to one acre (~4050 m$^2$). However, for five high-density residential classes ('APARTMENT HOUSE (100+ UNITS)', 'APARTMENT HOUSE (5+ UNITS)', 'APARTMENTS (GENERIC)', 'COOPERATIVE (RESIDENTIAL)', 'HIGHRISE APARTMENTS'), parcels of up to 10 acres (~40,500 m$^2$) were included in this "small" categorization. These thresholds were utilized to exclude parcels that contain large areas of open space in addition to residential structures. This was most commonly seen in RURAL RESIDENCE (AGRICULTURAL) parcels, which have a size (~18 acres) of roughly 40x that of average SINGLE FAMILY RESIDENTIAL parcels (~0.45 acres), on average, despite both types tending to encompass just one single family house. Therefore, parcel sizes for most residential parcel types were limited to one acre so that any open space contained within them would not exceed the size of a medium to large yard surrounding a single-family home ([S2 Fig]). This one-acre threshold corresponds to roughly 40% of the area of a single 100 x 100m grid cell and therefore, any open yard space within these plots are likely to minimally impact the eventual gridded output. The 10-acre threshold utilized for the five high-density parcel types was selected after manual inspection of those parcel classes were determined to often occupy more area (i.e. a full city block) in urban zones without containing large amounts of unpopulated space.

These small residential parcels were selected as target zones amongst the ancillary data because most inaccuracies observed upon manual inspection of several hundred parcels were instances of large open spaces being assigned a residential use code, masked out here by selecting only small residential parcels. Given the somewhat common occurrence of false negatives in the building footprint dataset, we did not want to rely on these footprints alone to constrain populated area, though other gridded population efforts have employed such an approach, including the "constrained" WorldPop population grids [28] and those produced by Huang et al. (2021) [25] for the contiguous U.S. (CONUS) region.

*Identified all building footprints within "large" residential parcels not included in step (A) and combined those footprints with the small residential polygon boundaries from (A) to produce the final target zones for populations within all census blocks containing some residential parcel area.*

Building footprints within all "large" residential parcels were assumed to be residential and selected as target zones for population, which masked out open space in these large parcels but still included likely housing structures. These building polygon geometries were then merged with the small residential parcel polygons selected in step (A) to produce the final target zone extents for population apportionment within each census block. Therefore, the target zones within a single census block could consist of both small residential parcel boundaries as well as building footprints if both small and large residential parcels are present.

Given the fact that intersecting the census block, residential parcel and building footprint polygon geometries resulted in some small slivers or fragments of individual parcels or building footprints being assigned to certain blocks, a sliver-removal algorithm was employed to remove most of these instances using an upper population density limit of 1 person per 10 m$^2$.

Slivers are defined as any single-part polygon resulting from the block-parcel intersection that is less than:

*[original residential parcel area] / [2 \* # of polygons descendent of a given parcel after intersecting with blocks]*

In other words, if a residential parcel with an area of 1km$^2$ is split evenly across two different blocks into two 0.5km$^2$ portions, they will both be preserved since 0.5km$^2$ > [1km$^2$ / (2 x 2) = 0.25km$^2$]. However, if this same parcel is split across two blocks such that 90% (0.9km$^2$) of its area is contained in one block and 10% (0.1km$^2$) in the other, the smaller portion would be considered a sliver and removed since it is less than 0.25km$^2$. This ultimately resulted in the removal of 3.1% of polygons (in terms of count, not area) resulting from the intersection of census blocks with residential parcels.

B) This threshold value roughly corresponds to the top 99.9th percentile of observed population density in the original census block source zones and was employed to remove erroneous target zone geometries composed only of small sliver/fragment polygons. Steps (A) and (B) were applied to all census blocks that had some amount of residential parcel area.

C) *Identified all building footprints within census blocks with a non-zero population, but which contain no residential parcels, and set as target zones for those blocks.*
A small portion of the state's population resides in census blocks without residential parcels, largely in sparsely-populated regions. In these instances, the only ancillary data available was the building footprint data and population was uniformly apportioned across all building geometries in these cases. The main limitation in this approach is that not all structures are residential, resulting in some likely over-apportionment of populations to non-residential structures, an issue that was largely avoided in step (B) by selecting only buildings within large residential parcels.

D) *Identified any remaining blocks that have a non-zero population but do not contain any residential parcels nor building footprints. We used the census block boundary as the target zone in this case, assuming uniform population distribution across these blocks.*
A tiny fraction of the state's estimated 2020 population are enumerated in census blocks with neither residential parcels nor detected building footprints. In these cases, the target zones were simply treated as equivalent to the source zones (census block boundaries) and populations were uniformly distributed throughout these areas.

The resultant target zone polygonal geometries produced in steps (A-D) were uniformly assigned population densities based on their parent source zone populations and then converted to 100m x 100m statewide raster grids, which contain values of people per pixel (Fig 2). Roughly 34.5 million people in this final grid (87.2% of the state population) fell within small residential parcels (Step A), 3.6 million (9.1%) resided in large residential parcels and therefore are represented by building footprints within those parcels (Step B), 1.3 million (3.4%) fell in blocks with no residential parcels identified and therefore uniformly represented across all building footprints within these blocks, and for 128,000 people (0.3%) there existed neither residential parcel boundaries nor building footprints within their census blocks, resulting in a uniform distribution of those populations across their entire block areas (Fig 3). All geospatial operations were performed in a PostgreSQL (v13.3) programming environment using the PostGIS (v2.5) spatial database extension.

## Comparison to other gridded products

Four different, commonly-used global gridded population products were evaluated against the CA-POP grids: i) Gridded Population of the World v4.11 (GPW), ii) WorldPop (100m,

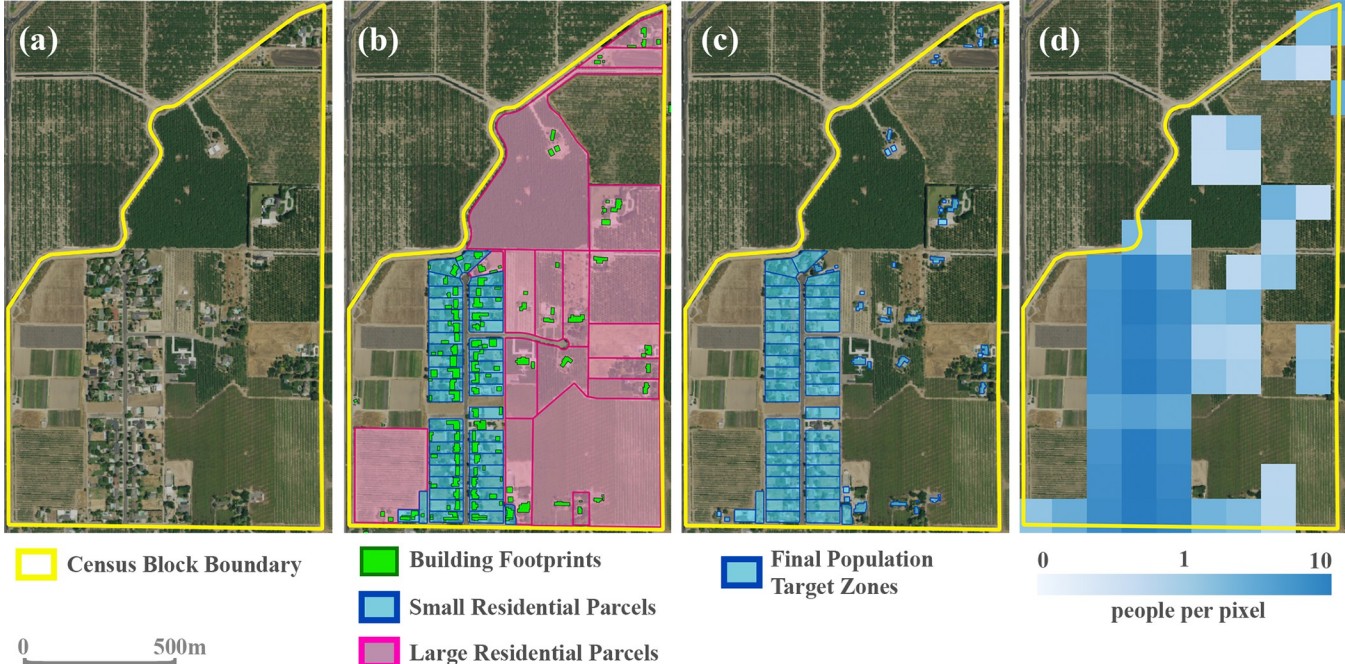

**Fig 2. CA-POP's dasymetric mapping method applied to a single census block.** Example shown is just north of Modesto, CA. Panel (a) shows the block boundary; (b) shows the ancillary residential parcel and building footprint boundaries; (c) shows the polygon boundaries used as the final target zones to assign the block's population values, retaining the small residential parcel polygons and building footprint polygons within large residential parcels; (d) shows the 100m-resolution grid produced from population apportioned to the final target zones. (Satellite base imagery source: USGS (NAIP) from The National Map).

unconstrained) (WP$_{UC}$), iii) WorldPop (100m, constrained) (WP$_C$), and iv) LandScan (LS). At the time of writing, each of these products' population source zones were based on the 2010 census at the block level, with populations in later-year grids estimated via different growth forecasting and/or inter-census population estimates to extrapolate 2010 values over time [28, 45, 46]. The latest GPW, WP$_{UC}$ and WP$_C$ population grids are for 2020 and LS for 2019, though presumably they will be updated to 2020 census block source zone populations in the near future. We also assessed two sets of population grids produced for the CONUS region by: i) Huang et al. (2021) [25], which utilized an earlier version of the Microsoft building footprints as ancillary data, and ii) the SocScape grids, which were produced using census block population estimates along with two land use ancillary datasets from 2010–2011 [20].

The GPW product employs the simplest methodology of the grids evaluated, apportioning population from source zones (blocks) to grid pixels through a simple, uniform areal weighting technique, masking out some unpopulated zones, such as water bodies and is provided at a 1km resolution. WorldPop employs a much more complex approach based on constructing machine learning models using a wide suite of covariates, such as roads, land cover, nighttime lights, infrastructure, protected areas, among others to predict population distributions within source zones [11, 28]. The WP$_C$ grids represent the results of these predictive models, but with population constrained to building footprints as represented in a recent buildings dataset from Maxar/Ecopia (WorldPop.org, [28]). Both WorldPop datasets are provided at 1km and 100m grid resolutions. LandScan employs a "smart interpolation" approach to weight pixels by likelihood of containing population based on a large suite of ancillary data and apportioning populations accordingly, and is provided at a 1km resolution [28, 46].

Therefore, one advantage of CA-POP over the GPW and LS grids is its higher resolution (100m compared to 1km), made possible from the fine scale parcel and building footprint

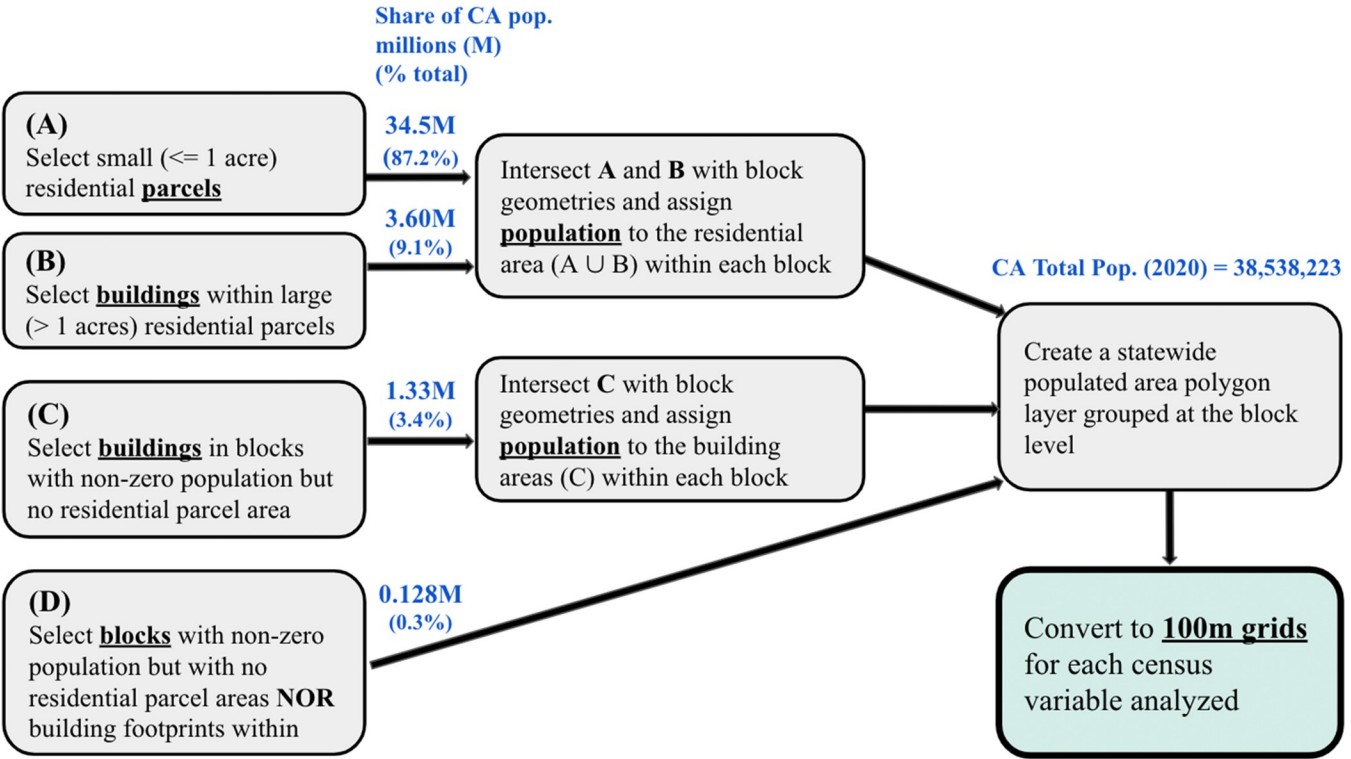

**Fig 3. Process workflow illustrating the identification of population target zones.** Dasymetric mapping process using the residential parcel and building footprint ancillary datasets within each census block and producing the final statewide grids for each population variable considered.

ancillary data utilized in its production. This allows for a more granular representation of population distributions, especially in sparsely populated regions (Fig 4). Also, the dasymetric mapping techniques used in CA-POP are a significant improvement over GPW's simple, areal weighting techniques that assume uniform population distribution throughout census blocks. The CA-POP techniques are simpler than the more heavily-modeled approaches used in LS

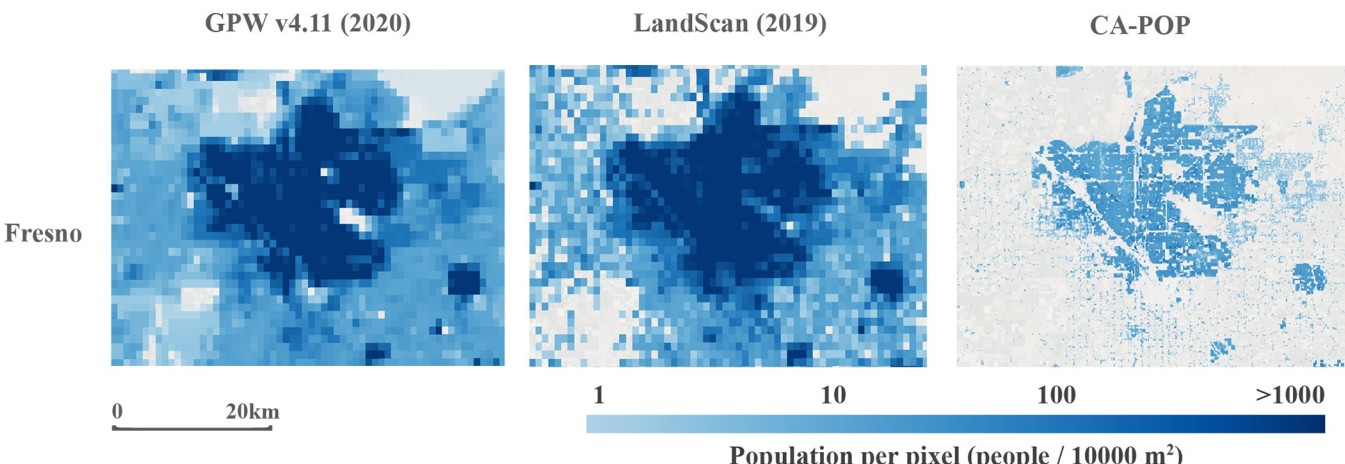

**Fig 4. CA-POP compared to the GPW and LS 1km resolution datasets.** Examples are shown for the Fresno area, CA. (Satellite base imagery source: USGS (NAIP) from The National Map).

and the WP grids, requiring fewer input datasets and are therefore more easily-reproducible. The WP and LS products are particularly well-suited for predicting populations in regions of the world where official population estimates are sparse or more coarsely resolved compared to U.S. census blocks.

The CONUS level grids produced by Huang et al. (2021) [25] represent the closest, previously published, methodological approach to CA-POP, apportioning population from census tract source zones to Microsoft building footprints (v1) that fall within residential areas identified in a national OpenStreetMap land use database. However, by assigning population solely to Microsoft building footprint boundaries, the approach is vulnerable to the building detection errors associated with that dataset, namely the occurrence of false negatives. Additionally, population values and source zones in the study correspond to 2017 American Community Survey census tracts, a much coarser geographic unit compared to census blocks. Also, compared to the CONUS grid produced by Huang et al. (2021) [25], which only provides population counts, CA-POP offers grids for multiple sociodemographic variables in addition to population.

The SocScape grids represent the finest resolution population grids we evaluated, with population grids for total population as well as various racial subgroups at a 30-meter resolution for the CONUS region. These grids were produced using census blocks as source zones and a pair of national land use/land cover datasets as the ancillary layers to comprise target zones of population apportionment within blocks [20]. These grids are publicly-available for download at socscape.edu.pl and appear to have been recently updated to include grids based on 2020 census values. Table 2 summarizes the characteristics of each these gridded datasets that were compared to CA-POP.

**Table 2. Description of gridded datasets assessed.** GPW, WPC, WPUC and LS datasets all currently use 2010 census blocks as their source zones, which will likely be updated to 2020 census blocks in subsequent grids.

| Product | Resolution | Population Apportionment Methods | Gridded Variables |
|---|---|---|---|
| Gridded Population of the World v4.11 (GPW) | 1km | Areal weighting assuming uniform population distribution using limited land use data (e.g. water bodies) to mask uninhabited areas | Population count |
| | | | Population density |
| | | | Population by age and sex (5-year age bins) |
| WorldPop (unconstrained) (WP$_{UC}$) | 100m | Random forest machine learning algorithm using a wide array of gridded and binary/categorical input covariates (e.g. topography, land cover, nighttime lights, local climate variables etc.) | Population count |
| | | | Population density |
| | | | Population by age and sex (5-year age bins) |
| WorldPop (constrained) (WP$_C$) | 100m | Similar modeling approach and input datasets as WP$_{UC}$ but with population limited to Maxar/Ecopia building footprint boundaries | Population count |
| | | | Population by age and sex (5-year age bins) |
| LandScan (LS) | 1km | "Smart interpolation" modeling approach to weight pixels based on their likelihood of containing population using a large suite of ancillary datasets (e.g. topography, land cover, climate, infrastructure etc.) | Population count |
| Huang et al. 2021 | 100m | Assigns population to Microsoft building footprints (v1) that are masked to residential areas using OpenStreetMap land use data using census tracts as source zones | Population count |
| SocScape 2010, 2020 | 30m | National Land Cover and Land Use Datasets as ancillary layers and census blocks as source zones | Population count Six racial subgroups |
| CA-POP | 100m | Uses high resolution statewide tax parcel dataset from LightBox-DMP and Microsoft building footprints (v2) as ancillary datasets to apportion populations from 2020 census block source zones | Population count (total) |
| | | | Hispanic/Latinx population |
| | | | Non-Hispanic/Latinx population for six racial subgroups and minors |

The incongruity of population source zone vintage between many of these alternative gridded products and CA-POP made direct comparisons to most of these grids challenging, and producing an earlier (i.e. 2010) version of CA-POP would be infeasible given the more recent vintage of its ancillary data. Additionally, once the census block source zone populations underpinning each of the other gridded products are updated to the 2020 census, block-level errors should in theory be zero and equal to those associated with the CA-POP grids. Given the fact that each of these grids utilize the highest resolution observed population estimates available, there is no straightforward way to estimate errors in their apportionment of population from source to target zones.

Therefore, we conducted a qualitative assessment of the advantages and disadvantages of CA-POP to the four global products based on their underlying data, methods and final grid resolutions, as well as through a series of manual accuracy assessments using contemporary satellite base-imagery from 2020–2021. However, reported metrics of relative errors as reported in the SocScape documentation were also compared to CA-POP, as the accuracy assessment employed by Dmowska and Stepinski (2017) [20] represents the most similar accuracy assessment to the one we employed for the CA-POP grids.

## Accuracy assessment

Census blocks are the finest spatial unit of population estimation tabulated in the census, and therefore represent the highest resolution set of "ground-truth" population estimates available to evaluate population estimation accuracy for different dasymetric modeling exercises. Given the fact that these block-level populations are used to constrain population totals in these grids, there is not an easy way to assess the relative accuracy of the dasymetric mapping approach aside from physically visiting the areas or manual inspection of the final product against recent satellite imagery and with contextual knowledge about likely populated areas. More highly-modeled, machine learning based prediction models of population can perform accuracy assessments using a cross-validation process, whereby certain observed population constraints are withheld from the modeling process and errors at those omitted locations relative to observed values are measured [30, 47]. However, past applications of traditional dasymetric mapping techniques, like those employed in this study, have often evaluated accuracy of their techniques by producing population grids that are constrained by observed populations at a spatial unit that is coarser (e.g. block groups or tracts) than the finest unit available, then evaluating how well the disaggregated population in the resultant grids matches observed population values within the finest spatial unit of ground-truth data (e.g. blocks) [15, 17, 25].

We performed this form of accuracy assessment for our mapping technique by producing a statewide grid using 2020 population estimates at the block-group level (~20x larger than blocks) but using the same ancillary datasets and dasymetric mapping process described above. Block-level errors were then calculated by comparing values from this grid within each census block's census population value. Generally, the highest percent errors at the block-level in this analysis occurred in blocks with low population totals, both due to the lack of residential parcel data within them and their small population values (i.e. small denominator in the percent error calculations) (S3 Fig). However, these errors are not exactly analogous to those that exist in the block-constrained grids comprising the final CA-POP products, which are likely lower in magnitude due to the finer resolution of the input data.

We assessed the relative accuracy of our methods compared to simple uniform, areal population weighting. The errors in each block-level estimate between the modeled (block group constrained grid) and observed (census block level data) are reported in terms of root mean-squared errors (RMSE) and the squared Pearson correlation coefficient ($R^2$) across all

populated blocks. Median block-wise percent errors were also calculated for both raw and absolute percent error magnitudes, following a similar accuracy analysis approach employed by Dmowska and Stepinski (2017) [20], in which they term the median absolute percent error a measure of 'relative error'. Given the skewed nature of the percent error distribution across population blocks, the median was deemed a more informative metric as opposed to the mean or standard variation [20].

The same procedure was carried out for a simple uniform, areal weighting technique, which estimates block level population values assuming a homogenous population distribution across the entire spatial area of each block group. Unfortunately, comparison of CA-POP grids produced using this 'second-best' ground-truth source zone population data (block-groups) to other products (e.g. WorldPop, LandScan) is not possible due to the fact that those data providers do not provide versions of their grids that utilize anything besides the best-available source zone data as their ground-truth population constraints (blocks). However, SocScape's documentation reports error estimates for a version of their 2010 population grid constrained by block groups instead of blocks, allowing for a roughly analogous error comparison between CA-POP and SocScape [20].

## Results

Statewide, 100-meter resolution raster grids were produced for each of the eight 2020 U.S. Census demographic variables listed in Table 1 for the entire state of California, comprising the CA-POP dataset. Examples of these grids for four demographic variables at different locations across the state are provided in Fig 5 and are publicly available online (see Data Availability).

Assessing the accuracy of the block group-constrained CA-POP grid to the simple uniform areal weighting technique demonstrates the improved accuracy of using these dasymetric mapping techniques, in terms of lower absolute error magnitudes (RMSE) and higher agreement of population distribution ($R^2$) at the block level compared to uniform areal weighting (Table 3). In terms of median percent errors, the CA-POP method yields much higher accuracy compared to the simple uniform method both in raw (-4.1% compared to -25.9%) and absolute terms, or 'relative error', (30.1% compared to 46.4%). This median relative error of 30.1% is also lower than the 44% relative error value reported for SocScape's 2010 national population grid, calculated using the accuracy assessment exercise of comparing block group-constrained grid performance to observed block values [20]. Although SocScape's relative error is a CONUS-wide value and CA-POP's value of is solely for California, making an analogous comparison impossible, CA-POP's lower error value suggests that it, on average, likely outperforms SocScape.

However, it is important to consider that these accuracy values only reflect the improved accuracy of the block group-constrained grid compared to uniform areal weighting, and not the block-constrained grid, which is how the final grids in this study were produced. By design, the block level errors of the block-constrained grid are zero, and calculating an analogous set of accuracy measures would require ground-truth estimates of population at the sub-block target zones (i.e. residential parcels and buildings), which are not available. Therefore, the accuracy improvements as compared to uniform areal weighting shown in Table 3 are simply to demonstrate the value of utilizing the dasymetric mapping techniques in CA-POP generally, and do not reflect exact accuracy values of the final, block-based grids, which by definition are more accurate across space than block group-based grids.

We also assessed pixel-level differences between the 2020 SocScape and CA-POP total population grids across the state to better evaluate how apportionment of population at the sub-

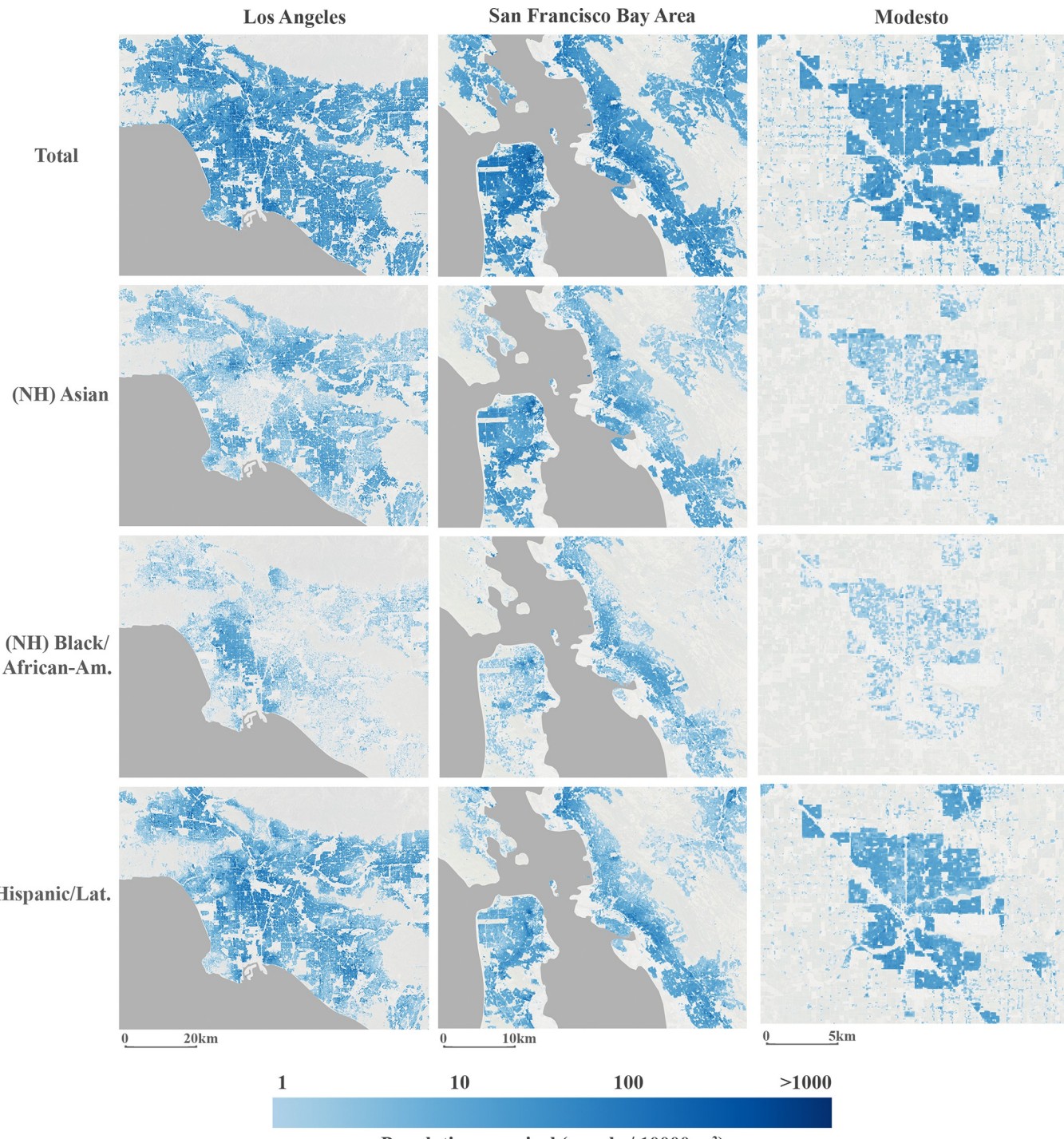

**Fig 5. Final CA-POP grids.** Examples are shown for four demographic population variables at three different locations in California. (Satellite base imagery source: USGS (NAIP) from The National Map, Ocean boundary layer source: Natural Earth).

block scale differs between the two methods. Fig 6 displays SocScape pixel values subtracted from CA-POP as a grid, and demonstrates that SocScape distributes low population counts across the majority of open space within blocks, whereas CA-POP more accurately sets these

**Table 3. Summary error statistics of block group-constrained CA-POP grid.**

| Block Population Estimation Method | RMSE (people) | $R^2$ | Median Percent Error | Median Absolute Percent Error (Relative Error) |
|---|---|---|---|---|
| Block group-constrained, dasymetric population grid (CA-POP method) | 76.9 | 0.76 | -4.1% | 30.1% |
| Uniform, areal weighting of block group population | 114.6 | 0.54 | -25.9% | 46.4% |

Differences between the block group constrained dasymetric population grids and simple, uniform areal population estimation techniques for populated blocks only (unpopulated blocks excluded).

regions to zero. This is evident in the figure's first two panels, where the light red zones spanning large areas represent regions in SocScape with small population counts across primarily open space, for which CA-POP does not apportion population. In more densely-populated urban zones where blocks are smaller and the two grids are therefore constrained to equal one another at a smaller spatial scale, much of the pixel-level differences emulate random noise, although some patterns appear to suggest that SocScape overly-apportions populations along major streets and roadways where CA-POP does not. The final panel in Fig 6 demonstrates this in South Los Angeles, where red areas (SocScape greater than CA-POP) tend to reflect the pattern of major streets in this neighborhood, a difference that is likely due to CA-POP's use of ancillary datasets that exclude street surfaces from the population target zones.

In comparing CA-POP to other gridded products, given the high resolution of the population source zones and ancillary datasets used in CA-POP, it is not apparent that WP and LS approaches produce more accurate grids than CA-POP in a California context. In fact, the global Maxar/Ecopia building footprint dataset used to constrain the population apportionment extents in $WP_C$ looks to poorly capture residential structures in many areas of California, especially in medium to low density settings, compared to the Microsoft building footprint data used in CA-POP (Fig 7). Conversely, the $WP_{UC}$, which utilizes nighttime lights and

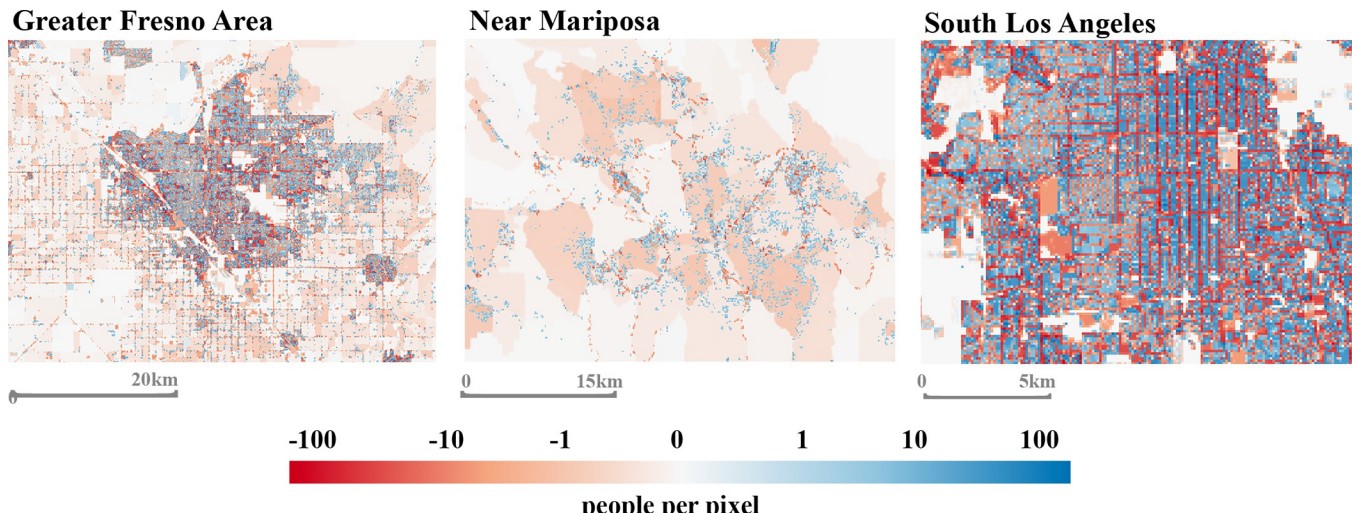

**Fig 6. Pixel-wise differences between the CA-POP and 2020 SocScape total population grids.** SocScape's 2020 total population grid was converted to population density, aggregated from 30m to 100m resolution using an average resampling approach and then re-converted to units of people per cell prior to differencing with CA-POP's total population grid. Blue areas represent regions where CA-POP pixel values are greater than SocScape and red areas are those where SocScape is greater.

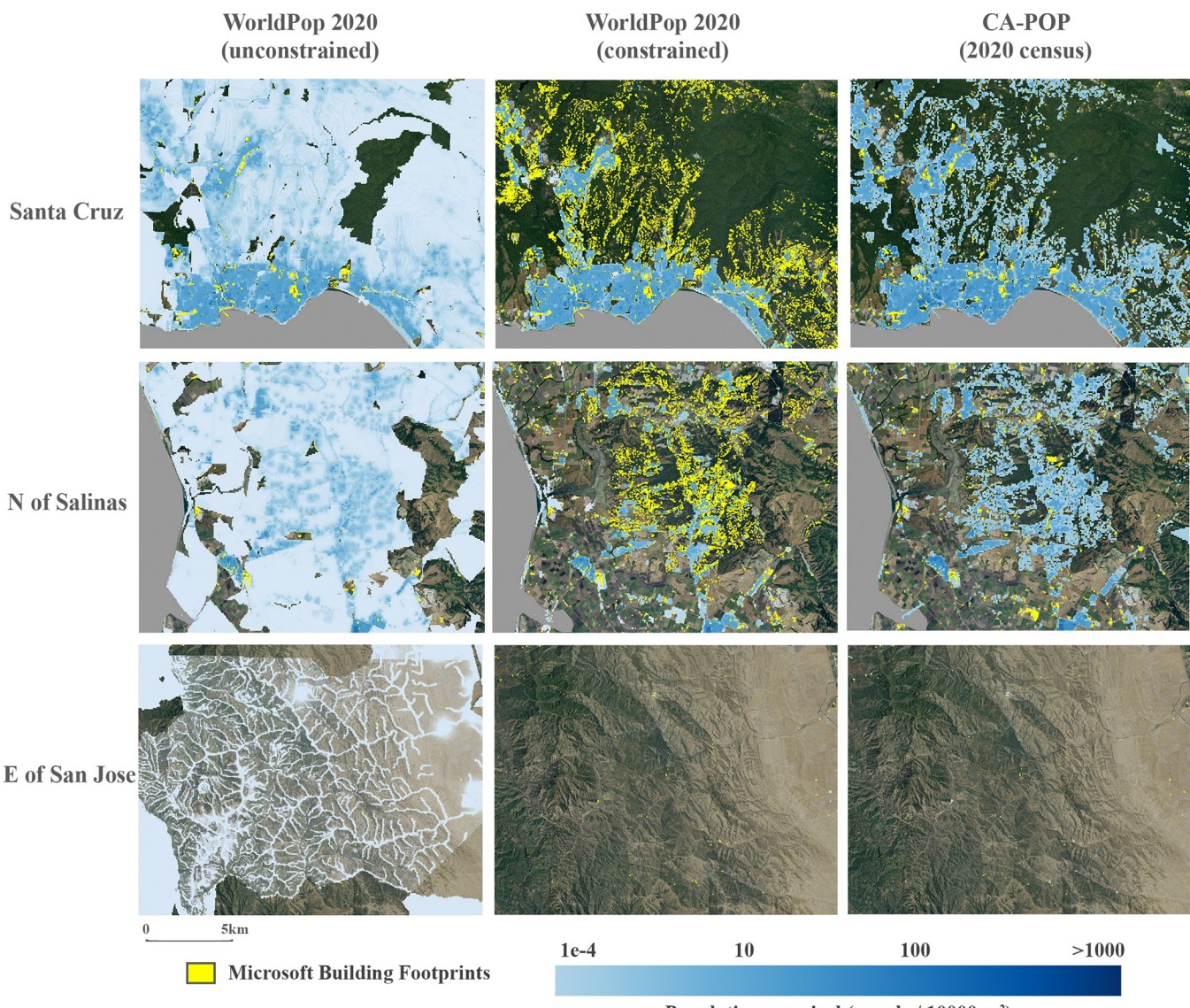

**Fig 7. Comparison of CA-POP with the unconstrained and constrained WorldPop grids.** Example shown for three locations in California. The first two rows represent populated areas and the third row represents largely unpopulated, open space. Over-apportionment of population across open space is seen in the unconstrained WorldPop grid and under-apportionment to buildings footprints detected by Microsoft (yellow area) in residential parcels is evident in the constrained WorldPop grid, as compared to CA-POP. (Satellite base imagery source: USGS (NAIP) from The National Map).

topography covariates in its predictive modeling algorithm seems to routinely assign population values to pixels based on the pattern of light scatter from street lights or topographic characteristics of the landscape, resulting in the allocation of low population densities across vast swaths of uninhabited space (Fig 7).

## Discussion

Overall, the CA-POP grids look to perform well when compared against other available gridded population products (e.g. GPW, WorldPop, LandScan, SocScape) in terms of its high resolution and ability to capture known residential areas in its population apportionment. CA-POP also contains additional demographic variables compared to many alternative

products, which can be of use in many research applications concerned with specific demographic subgroups, such as environmental health, equity and justice-oriented research. Gridded raster products allow for easier spatial analysis of values within a given zone of interest compared to vector polygon layers due to the relative ease of summing or averaging pixels within an area as opposed to intersecting multiple vector polygon layers and conducting some form of subsequent areal weighting within the zone of interest.

## Limitations and potential improvements

Though CA-POP represents a fairly accurate and easily-replicable method of gridding different census variables, there are a number of known limitations and improvements that could still be made. For one, the accuracy of the CA-POP grids relates to the certainty of the 2020 U. S. census values, which was unique for a number of reasons, including the COVID-19 pandemic, a potential citizenship question, natural disasters, and various operational changes to the census enumeration process that may have led to an undercount of particular groups, especially marginalized populations [48]. Recently, The Urban Institute approximated this possible bias by state and urban area through a process of statistically simulating the likely "true" census values, with their results suggesting that the 2020 census in California is biased low by roughly 345,097 people (-0.87%) [48]. Nationally, the study indicates that these undercounts are proportionally higher in certain population subgroups, such as Black and Hispanic communities and for young children (-2.45%, -2.17% -4.86%, respectively) [48]. Therefore, the CA-POP grids based on these census values should be interpreted with the knowledge that statewide totals are likely lower than true populations, especially in certain disadvantaged communities.

Other future improvements to CA-POP could include updating ancillary datasets as they are made available, such as residential tax parcel boundaries of a more recent vintage than 2017–2018, or more accurate building footprint data. In theory, perfectly accurate building footprint data could be used for all final populated target zone boundaries, with tax parcels or other land-use ancillary data layers solely used to identify residential zones within which buildings should be selected, therein avoiding minor inaccuracies associated with regions of open space within small residential parcels currently present in our methodology. Additional types of ancillary data could also be considered to further inform the apportionment of population to eventual target zones within census blocks, such as home address data [17, 23] or mobile phone usage data [26]. The ancillary building footprint data that is utilized can also be potentially further analyzed to infer additional information about likely building types based on patterns and characteristics of building geometry and proximity to one another, analyses for which Jochem and Tatem (2021) [49] constructed the R package *foot*.

Additionally, some form of multi-class weighting technique could be employed to apportion population between different residential parcel types, as is done in similar studies [14, 18, 22]. This would require estimating the different relative population density in each of the 30 residential parcel types and then distributing population within a single source zone according to those weights, rather than distributing population evenly across the target zones within each block. Also, additional demographic census variables to the eight initial CA-POP grids provided here are planned to be produced as they are released from the U.S. Census.

Finally, the methods we utilize here are inherently limited in geographic scope given that only California is represented. The feasibility of constructing a U.S.-wide product using these dasymetric methods, however, is limited by the absence of national, high-quality, publicly-available tax parcel data. Tax parcel data are instead disparately gathered, maintained and provided by different state and local agencies, with a freely-available nationwide product with harmonized land use classifications not currently available for public use [20, 50]. A number of

proprietary options are maintained by various data retailers, though licensing fees often make access cost-prohibitive. Tax parcel boundaries are valuable ancillary datasets in many societally-beneficial demographic research contexts and we believe a publicly-funded effort to generate a well-maintained and open-access, national tax parcel dataset should be initiated to help facilitate this work.

## Conclusion

In this study, we present a set of high-resolution gridded population products using values from the 2020 U.S. Census for the entire state of California, known as 'CA-POP'. These grids were produced via dasymetric techniques, using census blocks as the population source zones, with population estimates from the 2020 census redistricting Summary File (P.L. 94–171), and leveraging two high-resolution ancillary datasets (residential parcel boundaries and building footprints), to reapportion the estimated population distributions at the sub-block scale.

Assessing the accuracy the CA-POP dasymetric mapping methodology for a population grid constrained by block group census observations instead of blocks yielded a block-wise median absolute relative error of approximately 30% for block group-to-block disaggregation, which is lower than national error rates reported in the CONUS-wide SocScape grids, the product that reports the most analogous form of accuracy assessment for block group-to-block population disaggregation grids derived from U.S. census values. Additionally, given that the final CA-POP grids are not constrained by block groups, but by higher-resolution census block observations, they are likely even more accurate than their block group-constrained counterparts over a given region, though a proper error assessment of these final grids is not possible due to the absence of observational data at the sub-block scale.

The statewide CA-POP population grids are publicly-available at a 100-meter resolution for eight population variables of interest provided by the 2020 census: total population, Hispanic/Latinx population of any race, and non-Hispanic populations of: American Indian/Alaska Native, Asian, Black/African-American, Native Hawaiian and other Pacific Islander, White, other race or multiracial (two or more races), and residents under 18 years old (i.e. minors).

## Supporting information

**S1 Table. Land use codes from tax parcel dataset identified as residential.**
(DOCX)

**S1 Fig. False negatives in the Microsoft building footprint data.** Examples shown in urban and rural contexts. Locations were chosen based on the presence of false negatives and do not generally reflect the typical proportion of false negative instances around the state.
(DOCX)

**S2 Fig. Examples of 1-acre residential parcels.** 1-acre (~4050 km$^2$) was used as the upper area threshold for low-density residential parcels.
(DOCX)

**S3 Fig. Percent errors of block group-constrained CA-POP population grid relative to census block population observations.** Values represent the percent difference between block-level estimates from the block group-constrained CA-POP total population grid and census block population values. Note that these errors do not reflect the block-level errors of the final CA-POP grids themselves, as those were constrained by the block-level census observations, which by definition makes block-level errors zero. Errors in the final grids instead occur at the sub-block level of population apportionment, for which there are no ground-truth population

observations available for assessing CA-POP's ability to apportion population within blocks. (DOCX)

## Acknowledgments

We thank the members of UC Berkeley's Sustainability and Health Equities Lab for their valuable feedback, guidance and initial application of these datasets. Thanks also to Dr. Maggi Kelly for her encouragement and feedback.

## Author Contributions

**Conceptualization:** Nicholas J. Depsky.

**Data curation:** Nicholas J. Depsky.

**Formal analysis:** Nicholas J. Depsky.

**Funding acquisition:** Lara Cushing, Rachel Morello-Frosch.

**Investigation:** Nicholas J. Depsky.

**Methodology:** Nicholas J. Depsky.

**Project administration:** Nicholas J. Depsky.

**Resources:** Nicholas J. Depsky.

**Supervision:** Lara Cushing, Rachel Morello-Frosch.

**Validation:** Nicholas J. Depsky.

**Visualization:** Nicholas J. Depsky.

**Writing – original draft:** Nicholas J. Depsky.

**Writing – review & editing:** Lara Cushing, Rachel Morello-Frosch.

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
