## [Decision Letter · Decision Letter 0]

15 Mar 2022

PONE-D-22-01737High-Resolution Gridded Estimates of Population Sociodemographics from the 2020 Census in CaliforniaPLOS ONE

Dear Dr. DEPSKY,

Thank you for submitting your manuscript to PLOS ONE. After careful consideration, we feel that it has merit but does not fully meet PLOS ONE’s publication criteria as it currently stands. Therefore, we invite you to submit a revised version of the manuscript that addresses the points raised during the review process.

Specifically, provide additional clarifications on comparison of CA-POP with other gridded products (qualitative versus quantitative comparisons), accuracy assessment, clarity on the presentation, spatial and temporal characteristics of the data, and finally limitations. Please submit your revised manuscript by Apr 29 2022 11:59PM. If you will need more time than this to complete your revisions, please reply to this message or contact the journal office at plosone@plos.org. Please include the following items when submitting your revised manuscript:A rebuttal letter that responds to each point raised by the academic editor and reviewer(s). You should upload this letter as a separate file labeled 'Response to Reviewers'.A marked-up copy of your manuscript that highlights changes made to the original version. You should upload this as a separate file labeled 'Revised Manuscript with Track Changes'.An unmarked version of your revised paper without tracked changes. You should upload this as a separate file labeled 'Manuscript'.

We look forward to receiving your revised manuscript.

Kind regards,

Krishna Prasad Vadrevu, Ph.D

Academic Editor

PLOS ONE

Journal Requirements:

“This study was funded by the California Air Resources Board (# 18RD018- RM-F and NJD), the Strategic Growth Council (CCRP0022 - RM-F, NJD and LC) and  U.S. Environmental Protection Agency (#84003901 LC, RM-F and ND)”

“This study was funded by the California Air Resources Board (# 18RD018- RM-F and NJD), the Strategic Growth Council (CCRP0022 - RM-F, NJD and LC) and U.S. Environmental Protection Agency (#84003901 LC, RM-F and ND)”

“This study was funded by the California Air Resources Board (# 18RD018- RM-F and NJD), the Strategic Growth Council (CCRP0022 - RM-F, NJD and LC) and  U.S. Environmental Protection Agency (#84003901 LC, RM-F and ND)”

4. We note that Figure 1, 2, 4 and 6 in your submission contain map images which may be copyrighted. All PLOS content is published under the Creative Commons Attribution License (CC BY 4.0), which means that the manuscript, images, and Supporting Information files will be freely available online, and any third party is permitted to access, download, copy, distribute, and use these materials in any way, even commercially, with proper attribution. For these reasons, we cannot publish previously copyrighted maps or satellite images created using proprietary data, such as Google software (Google Maps, Street View, and Earth). For more information, see our copyright guidelines: http://journals.plos.org/plosone/s/licenses-and-copyright.

 a. You may seek permission from the original copyright holder of Figure 1, 2, 4 and 6 to publish the content specifically under the CC BY 4.0 license. 

Reviewers' comments:

Reviewer's Responses to Questions

**Comments to the Author**

1. Is the manuscript technically sound, and do the data support the conclusions?

Reviewer #1: Yes

Reviewer #2: Yes

Reviewer #3: Partly

2. Has the statistical analysis been performed appropriately and rigorously? 

Reviewer #1: Yes

Reviewer #2: Yes

Reviewer #3: No

3. Have the authors made all data underlying the findings in their manuscript fully available?

Reviewer #1: Yes

Reviewer #2: Yes

Reviewer #3: Yes

4. Is the manuscript presented in an intelligible fashion and written in standard English?

Reviewer #1: Yes

Reviewer #2: Yes

Reviewer #3: Yes

5. Review Comments to the Author

Reviewer #1: In the manuscript entitled “High-Resolution Gridded Estimates of Population Sociodemographics from the 2020 Census in California” the authors present the CA-POP dataset, a series of high-resolution population grids for the state of California based on the 2020 census. The dataset contains eight demographic variables and is publicly and freely available from Github. It constitutes a valuable contribution that can be a basis for future sociodemographic research regarding California.

The dasymetric method used to obtain the CA-POP dataset was carefully and appropriately chosen. This choice as well as the method itself are explained in a very understandable way, and the reader gets a good sense of what has been done in the workflow, what information is contained in the layers, and what potential and limitations are associated with them. The figures are very helpful to understand the method and the relation of the source and ancillary data, as well as to grasp the difference between CA-POP and a similar product, the WorldPop grids.

I do like the manuscript in its present form, but I also have some suggestions for improvement:

- Did or do the authors consider to extend this dataset to the entire US (like the other mentioned gridded products)? It seems that, with the method now in place, this would be a reasonable extension to make and would certainly yield a hugely valuable dataset. If there is an interesting answer to this question, it may be worth to add it to the discussion.

- I think its unfortunate that you didn’t include a different method (possibly a machine-learning method) in the accuracy assessment comparison. It’s not very surprising that the dasymetric method used by CA-POP does better than a null model, after all a lot of informative ancillary data goes into it. So for a fair comparison and for making the point the CA-POP’s rather parsimonious method is appropriate here, it would have been more convincing to include a different algorithm that works with the same data/information.

- From the comparison of CA-POP with other gridded products I understand that the CONUS grids by Huang et al. 2021 are the most similar to CA-POP. The main benefit of CA-POP that you name in this direct comparison is that it uses the residential parcels to offset some weakness of the Microsoft building footprints. However, a more significant advantage that I would see here (taking from Table 3) is that CA-POP offers all this detail on the population composition, whereas Huang et al. 202 only give population count?

- At the beginning of the Results section (ll. 351-354), could you make it a little clearer how the raster grids described here differ from the CA-POP grids?

Line comments:

- l. 157: please define the term ‘data vintage’

- l. 253: I think there should be no dash between “1-acre” (as opposed to l. 255 where it’s surely correct)

- l. 269: please identify what the CONUS region is

- l. 356: “these methods” don’t really have a reference in a preceeding sentence; please directly name them (Actually, the whole sentence sounds like it was moved here from a different context.)

- l. 365, 367: unnecessary repetition of “these dasymetric techniques”

- Figure 6 is included twice

Reviewer #2: The authors present a set of high-resolution gridded population products using values from the 2020 U.S. Census for the entire state of California (CA-POP). This is a very thorough and interesting analysis that makes a nice contribution to the growing literature on high resolution grided estimates of population. In particular, some statistic value should be added in Abstract to illuminate the accuracy and good concordance of CA-POP. While the methods (dasymetric methods, accuracy assessment), the paper is often unclear and hard to assess. Too many details are provided for methods that ought to be summarized more succinctly. The paper needs to clarify the methods and improve the flow overall.

Reviewer #3: The manuscript entitled “High-Resolution Gridded Estimates of Population Sociodemographics from the 2020 Census in California” produced population grids by apportioning census block population to 100-m grids based on California tax parcel data and Microsoft building footprint. This manuscript is well-written, but some issues need to be addressed before publication.

1. The author tries to compare CA-POP accuracy with the areal weighting of block group population to demonstrate its superior performance. However, simple areal weighting is known for its bad performance in population downscaling when compared with other methods. I’d suggest including more dasymetric mapping methods to enhance the comparison (e.g., dasymetric mapping based on commonly-used ancillary dataset such as imperviousness, road, etc.). Adding a spatial map showing the overestimation/underestimation percentages of each block for the proposed method and related discussions is recommended.

2. The author tries to compare the CA-POP grids with other gridded product (e.g., WorldPop, LandScan, GPW), but only in a qualitative way (A table comparing their spatial resolutions, population apportionment methods, and gridded variables.) due to the intrinsic limitations of those products. As the accuracy assessment does not reflect the true accuracy of the CA-POP grids, a quantitative comparison with other gridded products becomes more urgent. SocScape (http://www.socscape.edu.pl/) provides 30-m resolution population grid (also racial diversity grids) for year 2020 across the United States. Aggregating this dataset to 100-m and then directly compare to the CA-POP dataset could greatly enhance the comparison analysis. If this direct comparison is included, then there’s no need to include the supplemental Table A2, which shows a biased accuracy assessment comparison. (Conducting accuracy assessments for these gridded products at the block level does not hold true since they are using blocks as the source zones, and as the author has discussed, the spatial resolution would heavily impact the accuracy at block level). Also, if this quantitative comparison is included, the qualitative comparison (Table 3) is less important and should be moved to the introduction part (Introducing in detail about these products there).

3. Dasymetric mapping of population often choose nation-wide ancillary dataset with high temporal resolution, making the product available to a greater spatial and temporal extent. While this study relies heavily on California tax parcel dataset in 2017/2018, it has a relatively limited spatial and temporal application. This is a major limitation and should be discussed in the manuscript.

4. The author has described the process of removing large residential parcels by thresholds, and the remaining small residential parcels have relatively small open space, which should have less impact on the eventual gridded population output. Even if the impact of these open space is minimalized by only selecting small residential parcels, it should still be considered as a limitation for the proposed method, and I think it is still worth been mentioned and discussed for its potential solution in the “limitations and potential improvements” part. Another limitation is that the selection of the thresholds is based on manual inspection, which is considered as an impediment if this method is applied elsewhere.

5. “Residential parcels tended to be fairly homogenous within census blocks (i.e. a single block rarely contained both highrise apartments and single family homes or rural residences), reducing the need for a multi-class weighting scheme”. Reporting a detailed percentage value for this rare occasion in the study area could be more persuasive to the readers.

6. The conclusion part is week and should be enhanced.

There are some minor issues:

1. Page 3, line 68 – 70. The list of additional ancillary dataset is not exhaustive, and I recommend adding more variety of ancillary dataset:

Property data (Wan, H., Yoon, J., Srikrishnan, V., Daniel, B., Judi, D., 2021. Population downscaling using high-resolution, temporally-rich US property data. Cartography and Geographic Information Science 1–14);

Building footprint data (Huang, X., Wang, C., Li, Z., Ning, H., 2021. A 100 m population grid in the CONUS by disaggregating census data with open-source Microsoft building footprints. Big Earth Data 5, 112 – 113).

2. Page 12, line 286. Can the authors explain more in detail about the sliver-removal algorithm?

3. Page 15, line 338. The statement for evaluating errors should be clearer. The downscaled grid populations are first aggregated to the finest spatial unit (blocks), and then compared with the ground-truth observations at that spatial unit level.

4. Page 15 – 16, line 356 – 365. These sentences are more related to the accuracy assessment part rather than the result part.

5. Page 17, line 387. “The later-year grids are estimated via different growth forecasting assumptions to extrapolate 2010 values”. Are all those datasets extrapolating population for every year from 2010 to 2020? The author should be clearer about this statement.

6. PLOS authors have the option to publish the peer review history of their article (what does this mean?). If published, this will include your full peer review and any attached files.

Reviewer #1: No

Reviewer #2: No

Reviewer #3: No

---

## [Author Response · Author response to Decision Letter 0]

13 May 2022

Author responses to requested revisions to 

Depsky, Cushing, Morello-Frosch 2022: High-resolution gridded estimates of population sociodemographics from the 2020 census in California

[EDITOR] Journal Requirements:

We have formatted our manuscript in accordance with the above requirements.

“This study was funded by the California Air Resources Board (# 18RD018- RM-F and NJD), the Strategic Growth Council (CCRP0022 - RM-F, NJD and LC) and U.S. Environmental Protection Agency (#84003901 LC, RM-F and ND)”

We include an amended statement in the cover letter as follows: 

”This study was funded by the California Air Resources Board (# 18RD018- RM-F and NJD), the Strategic Growth Council (CCRP0022 - RM-F, NJD and LC) and U.S. Environmental Protection Agency (#84003901 LC, RM-F and ND). The funders had no role in study design, data collection and analysis, decision to publish, or preparation of the manuscript.”

“This study was funded by the California Air Resources Board (# 18RD018- RM-F and NJD), the Strategic Growth Council (CCRP0022 - RM-F, NJD and LC) and U.S. Environmental Protection Agency (#84003901 LC, RM-F and ND)”

“This study was funded by the California Air Resources Board (# 18RD018- RM-F and NJD), the Strategic Growth Council (CCRP0022 - RM-F, NJD and LC) and U.S. Environmental Protection Agency (#84003901 LC, RM-F and ND)”

We have removed all funding-related text from the manuscript and include an amended statement in the cover letter (See response to comment #1).

4. We note that Figure 1, 2, 4 and 6 in your submission contain map images which may be copyrighted. All PLOS content is published under the Creative Commons Attribution License (CC BY 4.0), which means that the manuscript, images, and Supporting Information files will be freely available online, and any third party is permitted to access, download, copy, distribute, and use these materials in any way, even commercially, with proper attribution. For these reasons, we cannot publish previously copyrighted maps or satellite images created using proprietary data, such as Google software (Google Maps, Street View, and Earth). For more information, see our copyright guidelines: http://journals.plos.org/plosone/s/licenses-and-copyright.

 a. You may seek permission from the original copyright holder of Figure 1, 2, 4 and 6 to publish the content specifically under the CC BY 4.0 license. 

We have now altered the map backgrounds in all map-based figures (Figs 1, 2, 4, 5, 6 and supplement figures S3_Fig and S4_Fig) to contain imagery from the USGS National Map Viewer and the ocean layer shown in Figures 4 and 6 is from Natural Earth. 

Each supplemental table or figure is now provided as a standalone file to be hyperlinked with corresponding captions at the conclusion of the manuscript immediately preceding the Data Availability section.

Reviewer #1

In the manuscript entitled “High-Resolution Gridded Estimates of Population Sociodemographics from the 2020 Census in California” the authors present the CA-POP dataset, a series of high-resolution population grids for the state of California based on the 2020 census. The dataset contains eight demographic variables and is publicly and freely available from Github. It constitutes a valuable contribution that can be a basis for future sociodemographic research regarding California.

The dasymetric method used to obtain the CA-POP dataset was carefully and appropriately chosen. This choice as well as the method itself are explained in a very understandable way, and the reader gets a good sense of what has been done in the workflow, what information is contained in the layers, and what potential and limitations are associated with them. The figures are very helpful to understand the method and the relation of the source and ancillary data, as well as to grasp the difference between CA-POP and a similar product, the WorldPop grids.

I do like the manuscript in its present form, but I also have some suggestions for improvement:

Did or do the authors consider to extend this dataset to the entire US (like the other mentioned gridded products)? It seems that, with the method now in place, this would be a reasonable extension to make and would certainly yield a hugely valuable dataset. If there is an interesting answer to this question, it may be worth to add it to the discussion.

Thank you for flagging this. We would have liked to have extended this analysis to cover the entire U.S. but were limited by the lack of contemporary, publicly-available tax parcel boundaries for each state. Given the patchwork manner in which parcels are assessed by state and local governments, there are no freely-available national tax boundary datasets with harmonized land use classes. Some data retailers offer parcel datasets with partial coverage of land use codes nationally (such as LoveLand Inc.), but they are offered as proprietary data products with licensing fees in the tens of thousands of dollars, which was out of the funding scope of this project and would prohibit the replicability and maintenance of the CA-POP grids with future census population values. We added an explanation of this at the end of the Discussion section (lines 591-600) as follows:

“Finally, the methods we utilize here are inherently limited in geographic scope given that only California is represented. The feasibility of constructing a U.S.-wide product using these dasymetric methods, however, is limited by the absence of national, high-quality, publicly-available tax parcel data. Tax parcel data are instead disparately gathered, maintained and provided by different state and local agencies, with a freely-available nationwide product with harmonized land use classifications not currently available for general use (Jia et al. 2014, Dmowska and Stepinski 2017). A number of proprietary options are maintained by various data retailers, though licensing fees often make access cost-prohibitive. Tax parcel boundaries are valuable ancillary datasets in many societally-beneficial demographic research contexts and we believe a publicly-funded effort to generate a well-maintained and open-access, national tax parcel dataset should be initiated to help facilitate this work.”

I think its unfortunate that you didn’t include a different method (possibly a machine-learning method) in the accuracy assessment comparison. It’s not very surprising that the dasymetric method used by CA-POP does better than a null model, after all a lot of informative ancillary data goes into it. So for a fair comparison and for making the point the CA-POP’s rather parsimonious method is appropriate here, it would have been more convincing to include a different (ML) algorithm that works with the same data/information.

We agree that comparing the accuracy of the CA-POP products to other population gridding products, such as those based on ML algorithms (e.g. WorldPop), would have been a compelling analysis. However, given the nature of these alternate population grids, such an analysis is not feasible. One reason is that to date most of the alternative gridded population products have not yet released grids based on the 2020 census values. Another primary reason such an analysis is not possible is the fact that most of the alternative major population grid examples (GPW, LandScan, WorldPop) all utilize the finest-scale ground-truthed population values available (census blocks) as their ‘source zones’ of population, just like CA-POP does. This is because when constructing dasymetric population maps, the most accurate end products will be those that utilize the best-available population observations, which in the U.S. are at the census block level from decennial censuses. Therefore, all of these products vary only in how populations are apportioned to subregions within each block, using sub-block ancillary data layers (e.g. building footprints or tax parcels) or via a ML-based population prediction algorithm, also trained using various ancillary data layers (e.g. nightlights, road networks etc.). When the alternate gridded products release their 2020-based population grids, there should essentially be no inaccuracies relative to census values at the block-level, as is the case with CA-POP. 

This is a common challenge when assessing the accuracy of dasymetric mapping techniques, and is described in Huang et al. 2021 and Stevens et al. 2015, because the desire to create dasymetric population maps is borne of a lack of ground-truthed population observations at the spatial resolution desired. However, that very lack of ground-truthed population data is precisely what prevents the ability to truly assess the accuracy of the method in a comprehensive, quantitative manner across the study space. Were there available ground-truthed data at the sub-block level, then such an assessment would be possible, but would also negate the need for creating a dasymetric map of downscaled population in the first place. Therefore, dasymetric mappers always face the dilemma of either foregoing the use of the highest-resolution ground-truthed population data available in favor of being able to conduct an accuracy assessment (e.g. using block-groups as population source zones of the final grid and then assessing its accuracy at the block-level), or using the highest-resolution ground-truthed data to construct a more accurate grid but therein forfeiting the ability to rigorously assess the accuracy given the lack of finer-scale observations.

Two scenarios would make a more convincing side-by-side accuracy comparison between CA-POP and other products like WorldPop or LandScan possible: i) for each product (including CA-POP) to create grids using block-groups, rather than blocks, as their population source zones and to make these publicly-available, allowing for an estimation of accuracy against ground-truthed block-level values; ii) the presence of some sub-block, ground-truthed population data across some part of California, such that the ability of CA-POP and other block-constrained products to disaggregate population within blocks could be benchmarked. However, to our knowledge neither of these requisite data products exist, making such an assessment infeasible. We do partially carry out the first option above, producing CA-POP grids using block-group source zones rather than blocks, but are only able to compare this to the null model (uniform, areal weighting) as this is all that is available for such a comparison.

The above explanation of these limitations is now summarized in the ‘Accuracy Assessment’ subsection of the Data and Methods section (lines 451-469), which also contains an additional concluding sentence detailing the barriers to conducting side-by-side comparisons with other gridded products:

“We assessed the relative accuracy of our methods compared to simple uniform, areal population weighting. The errors in each block-level estimate between the modeled (block group constrained grid) and observed (census block level data) are reported in terms of root mean-squared errors (RMSE) and the squared Pearson correlation coefficient (R2) across all populated blocks. Median block-wise percent errors were also calculated for both raw and absolute percent error magnitudes, following a similar accuracy analysis approach employed by Dmowska and Stepinski (2017), in which they term the median absolute percent error a measure of ‘relative error’. Given the skewed nature of the percent error distribution across population blocks, the median was deemed a more informative metric as opposed to the mean or standard variation (Dmowska and Stepinksi 2017). 

The same procedure was carried out for a simple uniform, areal weighting technique, which estimates block level population values assuming a homogenous population distribution across the entire spatial area of each block group. Unfortunately, comparison of CA-POP grids produced using this ‘second-best’ ground-truth source zone population data (block-groups) to other products (e.g. WorldPop, LandScan) is not possible due to the fact that those data providers to not provide versions of their grids that utilize anything besides the best-available source zone data as their ground-truth population constraints (blocks). However, SocScape’s methods documentation reports error estimates for their 2010 grids using the same type of accuracy assessment we carried out with CA-POP, constructing a block group-based grid and then comparing its ability to match observed population totals within blocks. This allows for a roughly analogous error comparison between CA-POP and SocScape (Dmowska and Stepinski 2017).”

From the comparison of CA-POP with other gridded products I understand that the CONUS grids by Huang et al. 2021 are the most similar to CA-POP. The main benefit of CA-POP that you name in this direct comparison is that it uses the residential parcels to offset some weakness of the Microsoft building footprints. However, a more significant advantage that I would see here (taking from Table 3) is that CA-POP offers all this detail on the population composition, whereas Huang et al. 202 only give population count?

We agree, thank you for this and added a sentence highlighting this on lines 395-397:

“Also, compared to the CONUS grid produced by Huang et al. (2021), which only provides population counts, CA-POP offers grids for multiple sociodemographic variables in addition to population.”

At the beginning of the Results section (ll. 351-354), could you make it a little clearer how the raster grids described here differ from the CA-POP grids?

These actually are the CA-POP raster grids, but our apologies that this was unclear. We added a “, comprising the CA-POP dataset” clause to the first sentence of the Results section to add clarity (lines 472-473), which now reads:

“Statewide, 100-meter resolution raster grids were produced for each of the eight 2020 U.S. Census demographic variables listed in Table 1 for the entire state of California, comprising the CA-POP dataset.”

Line comments:

157: please define the term ‘data vintage’

253: I think there should be no dash between “1-acre” (as opposed to l. 255 where it’s surely correct)

269: please identify what the CONUS region is

356: “these methods” don’t really have a reference in a preceeding sentence; please directly name them (Actually, the whole sentence sounds like it was moved here from a different context.)

365, 367: unnecessary repetition of “these dasymetric techniques”

Figure 6 is included twice

These were all implemented in the locations specified, thank you for the suggestions.

Reviewer #2

The authors present a set of high-resolution gridded population products using values from the 2020 U.S. Census for the entire state of California (CA-POP). This is a very thorough and interesting analysis that makes a nice contribution to the growing literature on high resolution gridded estimates of population. In particular, some statistical value should be added in Abstract to illuminate the accuracy and good concordance of CA-POP. 

We thank the reviewer for the suggestion. We have expanded the abstract to contain a brief discussion of our accuracy assessment, and include the 30% median relative error value as discussed in the text (see lines 26-34):

“A general accuracy assessment of the CA-POP dasymetric mapping methodology was conducted by producing a population grid that was constrained by block group census observations instead of blocks, enabling a comparison of this grid’s apportionment of population within census blocks to block-level census values. This accuracy assessment yielded a block-wise median absolute relative error of approximately 30% for block group-to-block disaggregation. However, the final CA-POP grids are constrained by higher-resolution census block-level observations, likely making them even more accurate than these block group-constrained grids over a given region, but for which error assessments of population disaggregation is not possible due to the absence of observational data at the sub-block scale.”

While the methods (dasymetric methods, accuracy assessment), the paper is often unclear and hard to assess. Too many details are provided for methods that ought to be summarized more succinctly. The paper needs to clarify the methods and improve the flow overall.

In response to specific requests for clarifications from other reviewers, we have addressed this in our revised manuscript, particularly in the methods section and accuracy assessment sections. revisions. Specifically, a number of additional clarifying statements have been added and the accuracy assessment portions of the Discussion and Results sections have been condensed for clarity and to reduce redundancies. We also placed some of the ‘Comparison to Other Gridded Products’ subsection text in the Results section to the Data and Methods section, as it more appropriately fits there and we believe should help with the manuscript’s flow and interpretability.

Reviewer #3

The manuscript entitled “High-Resolution Gridded Estimates of Population Sociodemographics from the 2020 Census in California” produced population grids by apportioning census block population to 100-m grids based on California tax parcel data and Microsoft building footprint. This manuscript is well-written, but some issues need to be addressed before publication.

The author tries to compare CA-POP accuracy with the areal weighting of block group population to demonstrate its superior performance. However, simple areal weighting is known for its bad performance in population downscaling when compared with other methods. I’d suggest including more dasymetric mapping methods to enhance the comparison (e.g., dasymetric mapping based on commonly-used ancillary dataset such as imperviousness, road, etc.). Adding a spatial map showing the overestimation/underestimation percentages of each block for the proposed method and related discussions is recommended.

Thank you for this feedback and we agree that simple areal weighting is not the most compelling comparison due it being the simplest option. Comparing CA-POP to other, more complex, methods would have been desired if such comparisons were feasible. We were not entirely clear if your request for “including more dasymetric mapping methods to enhance the comparison” is asking for us to either i) create various additional statewide dasymetric maps of population using different combinations of ancillary data ourselves, or to ii) compare CA-POP to other such dasymetric mapping products that have already been developed that rely on different ancillary data (e.g. WorldPop, LandScan, SocScape etc.). 

If this request is asking for the former, we contend that this would be out of the scope for this paper, as constructing multiple additional statewide grids, each from different sets of ancillary data, would multiply the scope of work for this analysis N-times by however many N-additional grids we create. Additionally, the methods behind the creation of each of these grids, intentionally constructed with lower resolution/precision ancillary data to those layers used in CA-POP, would also have to be fully documented in detail and would greatly expand the scope and length of this manuscript. 

If, however, the request is referring to the latter option, such that you would like us to compare CA-POP to other, previously-constructed gridded population products that utilize alternate dasymetric methods and ancillary datasets, this is a similar comment to the second comment from Reviewer 1 and would refer you to the response we provided above.

Regarding the request for ‘spatial map showing the overestimation/underestimation percentages of each block for the proposed method’, this would by definition yield a homogenous map of 0% errors at every block with respect to the final CA-POP grids, due to the fact that they are constrained by block-level population census observations. The errors in the final grids only occur at the sub-block apportionment of population to the smaller ancillary data footprints, for which we do not have ground-truthed observations against which to assess accuracy. The only feasible alternative was to produce a map of block-level errors with respect to the block group-constrained population grid that we used to assess general (upper-bounds) of the errors in the CA-POP method in apportioning from block-groups to blocks. We have therefore produced this map and provided a supplemental figure (S5_Fig) which portrays these block-level error percentages of the block group-constrained CA-POP for three locales. 

However, we did not include this figure in the main manuscript text because we feel it is potentially slightly misleading to readers, as it could be interpreted at first glance as errors of the final CA-POP grid values associated with each block. However, this error map instead represents different errors at blocks when applying our dasymetric method to coarser spatial units (block groups) of ground-truthed population observations, which is not in fact how the final grids were produced. However, given that this remains the best and only real way to gauge some relative accuracy of our methods (see our response to R1’s second comment), we agree that it provides utility and therefore include it in our supporting information and reference it in the manuscript text in lines 445-447, as follows:

“Generally, the highest percent errors at the block-level in this analysis occurred in blocks with low population totals, both due to the lack of residential parcel data within these and the small populations used as the denominator in these percentage error calculations (Figure in S5 Figure).”

The author tries to compare the CA-POP grids with other gridded products (e.g., WorldPop, LandScan, GPW), but only in a qualitative way (A table comparing their spatial resolutions, population apportionment methods, and gridded variables.) due to the intrinsic limitations of those products. As the accuracy assessment does not reflect the true accuracy of the CA-POP grids, a quantitative comparison with other gridded products becomes more urgent. SocScape (http://www.socscape.edu.pl/) provides 30-m resolution population grid (also racial diversity grids) for year 2020 across the United States. Aggregating this dataset to 100-m and then directly compare to the CA-POP dataset could greatly enhance the comparison analysis. If this direct comparison is included, then there’s no need to include the supplemental Table A2, which shows a biased accuracy assessment comparison. (Conducting accuracy assessments for these gridded products at the block level does not hold true since they are using blocks as the source zones, and as the author has discussed, the spatial resolution would heavily impact the accuracy at block level). Also, if this quantitative comparison is included, the qualitative comparison (Table 3) is less important and should be moved to the introduction part (Introducing in detail about these products there).

Thank you for pointing us towards the SocScape data products, we were not previously aware of them. We have added citations to the underlying methods paper for the SocScape data (Dmowska and Stepinski 2017) to various locations of our manuscript. We have also added two additional error statistics in our comparison of the block group-constrained CA-POP population grid to the uniform areal weighting block group-constrained grid of California. These block-wise accuracy metrics are the median percent error (raw) and median absolute percent error (called ‘relative error’ in Dmowska and Stepinski 2017). We found that this ‘relative error’ metric is roughly 30% for the block group-constrained CA-POP grid, compared to over 46% when using simple uniform, areal weighting. Dmowska and Stepinski report their CONUS-wide ‘relative errors’ based on block group-constrained grids to be around 44%, which suggests that CA-POP may have superior performance, at least when compared to the national average of SocScape (we added explanation of this in lines 483-491):

“In terms of median percent errors, the CA-POP method yields much higher accuracy compared to the simple uniform method both in raw (-4.1% compared to -25.9%) and absolute terms, or ‘relative error’, (30.1% compared to 46.4%). This median relative error is significantly lower than the 44% value of the equivalent metric reported for SocScape’s 2010 national population grid, calculated using the accuracy assessment exercise of comparing block group-constrained grid performance to observed block values (Dmowska and Stepinski 2017). Though the SocScape relative error is a national value and CA-POP’s is solely for California, it suggests that the CA-POP approach likely outperforms SocScape in this context.”

We appreciate your suggestion to compare CA-POP to the 30m SocScape grids. We downloaded the 2020 SocScape population grids and aggregated to 100m as suggested and compared the pixel-wise differences between CA-POP and SocScape across the state. This was helpful to evaluate the differences between the underlying dasymetric methods utilized in each approach. However, we feel it is important to note that the SocScape data are also products of dasymetric mapping techniques that disaggregate population from the block-level to sub-block target zones and are not ground-truthed observation data and therefore cannot be used to evaluate the accuracy of CA-POP in apportioning population within blocks. Based on the limited comparison of ‘relative error’ metrics from the block group-constrained CA-POP and (national) SocScape grids, CA-POP appears to perform better (30% error), at least relative to national performance of SocScape (44% error) reported in Dmowska and Stepinski 2017. The comparison of CA-POP to SocScape, however, was helpful in visualizing the differences between these grids and is described in Figure 6 and in lines 507-520:

“We also assessed pixel-level differences between the 2020 SocScape and CA-POP total population grids across the state to better evaluate how apportionment of population at the sub-block scale differs between the two methods. Figure 6 displays SocScape pixel values subtracted from CA-POP as a grid, and demonstrates that SocScape distributes low population counts across the majority of open space within blocks, whereas CA-POP more accurately sets these regions to zero. This is evident in the figure’s first two panels, where the light red zones spanning large areas represent regions in SocScape with small population counts across primarily open space, for which CA-POP does not apportion population. In more densely-populated urban zones where blocks are smaller and the two grids are therefore constrained to equal one another at a smaller spatial scale, much of the pixel-level differences emulate random noise, although some patterns appear to suggest that SocScape overly-apportions populations along major streets and roadways where CA-POP does not. The final panel in Figure 6 shows an example of this in South Los Angeles, where red areas (SocScape greater than CA-POP) tend to reflect the pattern of major streets in this neighborhood, a difference that is likely due to CA-POP’s use of ancillary datasets that exclude street surfaces from the population target zones.”

Dasymetric mapping of population often choose nation-wide ancillary dataset with high temporal resolution, making the product available to a greater spatial and temporal extent. While this study relies heavily on California tax parcel dataset in 2017/2018, it has a relatively limited spatial and temporal application. This is a major limitation and should be discussed in the manuscript.

Yes, thank you for flagging this. We would have liked to have extended this analysis to cover the entire U.S. but were limited by the lack of contemporary, publicly-available tax parcel boundaries for each state. Given the patchwork manner in which parcels are assessed by state and local governments, there are no freely-available national tax boundary datasets with harmonized land use classes. Some data retailers offer parcel datasets with partial coverage of land use codes nationally (such as LoveLand Inc.), but they are offered as proprietary data products with licensing fees in the tens of thousands of dollars, which was out of the funding scope of this project and would prohibit the replicability and maintenance of the CA-POP grids with future census population values. We have added an explanation of this at the end of the Discussion section (lines 591-600):

“Finally, the methods we utilize here are inherently limited in geographic scope given that only California is represented. The feasibility of constructing a U.S.-wide product using these dasymetric methods, however, is limited by the absence of national, high-quality, publicly-available tax parcel data. Tax parcel data are instead disparately gathered, maintained and provided by different state and local agencies, with a freely-available nationwide product with harmonized land use classifications not currently available for general use (Jia et al. 2014, Dmowska and Stepinski 2017). A number of proprietary options are maintained by various data retailers, though licensing fees often make access cost-prohibitive. Tax parcel boundaries are valuable ancillary datasets in many societally-beneficial demographic research contexts and we believe a publicly-funded effort to generate a well-maintained and open-access, national tax parcel dataset should be initiated to help facilitate this work.”

In terms of temporal limitations of the 2017-2018 parcel data used, we agree that ideally this would have more contemporary with the 2020 census data utilized and should be updated where possible with future updates to the CA-POP dataset. However, the 2017/2018 data were those to which we were granted access at the time of analysis.

The author has described the process of removing large residential parcels by thresholds, and the remaining small residential parcels have relatively small open space, which should have less impact on the eventual gridded population output. Even if the impact of these open space is minimalized by only selecting small residential parcels, it should still be considered as a limitation for the proposed method, and I think it is still worth been mentioned and discussed for its potential solution in the “limitations and potential improvements” part. Another limitation is that the selection of the thresholds is based on manual inspection, which is considered as an impediment if this method is applied elsewhere.

We agree and have added a clarifying sentence about this limitation in lines 572-576:

“In theory, perfectly accurate building footprint data could be used for all final populated target zone boundaries, with tax parcels or other land-use ancillary data layers solely used to identify residential zones within which buildings should be selected, therein avoiding minor inaccuracies associated with regions of open space within small residential parcels currently present in our methodology.”

Regarding the thresholds we utilized for inclusion of small residential parcels as population target zones, it’s true that their selection was guided by manual inspection of the parcel data around the state. However, we believe that the threshold of 1-acre for small residential lots would be suitable for other 100m population gridding exercises in different states or locales in which tax parcels are being utilized without the need for a new iteration of manual inspection. This is due to the fact that the 1-acre threshold for small residential parcels is sufficiently small in the context of a 100m raster grid (~40% of a single pixel’s area) and would still be large enough to encompass single family homes on small to moderate lots in much of the country.

“Residential parcels tended to be fairly homogenous within census blocks (i.e. a single block rarely contained both highrise apartments and single family homes or rural residences), reducing the need for a multi-class weighting scheme”. Reporting a detailed percentage value for this rare occasion in the study area could be more persuasive to the readers.

We agree with this assessment and we have opted to remove this justification argument from the manuscript text and instead highlight the lack of a multi-class weighting scheme as a limitation of the current methodological approach and potential area of future improvement. We made this decision because properly quantifying the degree of homogeneity of parcel types would entail determining mean population densities associated with all 30 residential parcel types across the state, requiring more complex geospatial analysis that is beyond the scope of this work (and would be the bulk of work required to carry out the more complex multi-class weighting scheme itself, since in quantifying this claim we would have to determine the weights of each land use class in order to identify exactly how low/medium/high density each is and compare diversity within all blocks). 

The conclusion part is weak and should be enhanced.

Additional summary of the accuracy assessment was added here (lines 609-617) to round out the conclusion and some reorganization was done to improve the flow of the section:

“Assessing the accuracy the CA-POP dasymetric mapping methodology for a population grid constrained by block group census observations instead of blocks yielded a block-wise median absolute relative error of approximately 30% for block group-to-block disaggregation, which is lower than national error rates reported in the CONUS-wide SocScape grids, the product that reports the most analogous form of accuracy assessment for block group-to-block population disaggregation grids derived from U.S. census values. Given that the final CA-POP grids are constrained by higher-resolution census block-level observations, they are likely more even more accurate than their block group-constrained counterparts over a given region, though a proper error assessment of them is not possible due to the absence of observational data at the sub-block scale.”

Line Comments:

Page 3, line 68 – 70. The list of additional ancillary dataset is not exhaustive, and I recommend adding more variety of ancillary dataset:

Property data (Wan, H., Yoon, J., Srikrishnan, V., Daniel, B., Judi, D., 2021. Population downscaling using high-resolution, temporally-rich US property data. Cartography and Geographic Information Science 1–14);

This citation was added as suggested in line 81

Building footprint data (Huang, X., Wang, C., Li, Z., Ning, H., 2021. A 100 m population grid in the CONUS by disaggregating census data with open-source Microsoft building footprints. Big Earth Data 5, 112 – 113).

This citation was added as suggested in line 81

“Page 12, line 286. Can the authors explain more in detail about the sliver-removal algorithm?

We provided additional explanation in the form of a footnote linked to line 298 as follows:

“Slivers are defined as any single-part polygon resulting from the block-parcel intersection that is less than:

[original residential parcel area] / [2 * # of polygons descendent of a given parcel after intersecting with blocks] 

In other words, if a residential parcel with an area of 1km2 is split evenly across two different blocks into two 0.5km2 portions, they will both be preserved since 0.5km2 > [1km2 / (2 x 2) = 0.25km2]. However, if this same parcel is split across two blocks such that 90% (0.9km2) of its area is contained in one block and 10% (0.1km2) in the other, the smaller portion would be considered a sliver and removed since it is less than 0.25km2. This ultimately resulted in the removal of 3.1% of polygons (in terms of count, not area) resulting from the intersection of census blocks with residential parcels.”

Page 15, line 338. The statement for evaluating errors should be clearer. The downscaled grid populations are first aggregated to the finest spatial unit (blocks), and then compared with the ground-truth observations at that spatial unit level.

We have altered this sentence (lines 433-438) to provide more clarity about this process as suggested as follows:

“However, past applications of traditional dasymetric mapping techniques, like those employed in this study, have often evaluated accuracy of their techniques by producing population grids that are constrained by observed populations at a spatial unit that is coarser (e.g. block groups or tracts) than the finest unit available, then evaluating how well the disaggregated population in the resultant grids matches observed population values within the finest spatial unit of ground-truthed data (e.g. blocks)”

Page 15 – 16, line 356 – 365. These sentences are more related to the accuracy assessment part rather than the result part.

We have placed most of this text in the Discussion-Accuracy Assessment section rather than results and consolidated the text to reduce redundancy, as suggested. The Results now simply highlight the accuracy values themselves, as shown in Table 2, rather than discussing methods. (now lines 451-459):

“We assessed the relative accuracy of our methods compared to simple uniform, areal population weighting. The errors in each block-level estimate between the modeled (block group constrained grid) and observed (census block level data) are reported in terms of root mean-squared errors (RMSE) and the squared Pearson correlation coefficient (R2) across all populated blocks. Median block-wise percent errors were also calculated for both raw and absolute percent error magnitudes, following a similar accuracy analysis approach employed by Dmowska and Stepinski (2017), in which they term the median absolute percent error a measure of ‘relative error’. Given the skewed nature of the percent error distribution across population blocks, the median was deemed a more informative metric as opposed to the mean or standard variation (Dmowska and Stepinksi 2017). The same procedure was carried out for a simple uniform, areal weighting technique, which estimates block level population values assuming a homogenous population distribution across the entire spatial area of each block group. Unfortunately, comparison of CA-POP grids produced using this ‘second-best’ ground-truth source zone population data (block-groups) to other products (e.g. WorldPop, LandScan) is not possible due to the fact that those data providers to not provide versions of their grids that utilize anything besides the best-available source zone data as their ground-truth population constraints (blocks). However, SocScape’s methods documentation reports error estimates for their 2010 grids using the same type of accuracy assessment we carried out with CA-POP, constructing a block group-based grid and then comparing its ability to match observed population totals within blocks. This allows for a roughly analogous error comparison between CA-POP and SocScape (Dmowska and Stepinski 2017).”

Page 17, line 387. “The later-year grids are estimated via different growth forecasting assumptions to extrapolate 2010 values”. Are all those datasets extrapolating population for every year from 2010 to 2020? The author should be clearer about this statement.

Yes, GPW (Doxsey-Whitfield et al. 2015) and WorldPop (Lloyd et al. 2019) detail the population growth extrapolation mechanisms in the non-decennial-year gridded products. The LandScan documentation (Rose et al. 2020) is slightly more opaque in this regard but does make mention of utilizing mid-year population estimates to adjust values. Therefore, we have altered the sentence in our manuscript slightly to more generally describe these varied methods (lines 350-353):

“At the time of writing, each of these products’ population source zones were based on the 2010 census at the block level, with populations in later-year grids estimated via different growth forecasting and/or inter-census population estimates to extrapolate 2010 values forward (Doxsey-Whitfield et al., 2015; Lloyd et al., 2019; Rose et al., 2020).”

However, we do not believe that describing the specific population extrapolation formulas utilized in these other products in finer detail is not required for the narrative of our text in this section.

---

## [Decision Letter · Decision Letter 1]

17 Jun 2022

High-resolution gridded estimates of population sociodemographics from the 2020 census in California

PONE-D-22-01737R1

Dear Dr. DEPSKY,

We’re pleased to inform you that your manuscript has been judged scientifically suitable for publication and will be formally accepted for publication once it meets all outstanding technical requirements.

Kind regards,

Krishna Prasad Vadrevu, Ph.D

Academic Editor

PLOS ONE

Additional Editor Comments (optional):

Reviewers' comments:

Reviewer's Responses to Questions

**Comments to the Author**

1. If the authors have adequately addressed your comments raised in a previous round of review and you feel that this manuscript is now acceptable for publication, you may indicate that here to bypass the “Comments to the Author” section, enter your conflict of interest statement in the “Confidential to Editor” section, and submit your "Accept" recommendation.

Reviewer #1: All comments have been addressed

Reviewer #2: All comments have been addressed

Reviewer #3: All comments have been addressed

2. Is the manuscript technically sound, and do the data support the conclusions?

Reviewer #1: Yes

Reviewer #2: Yes

Reviewer #3: Yes

3. Has the statistical analysis been performed appropriately and rigorously? 

Reviewer #1: Yes

Reviewer #2: Yes

Reviewer #3: Yes

4. Have the authors made all data underlying the findings in their manuscript fully available?

Reviewer #1: Yes

Reviewer #2: Yes

Reviewer #3: Yes

5. Is the manuscript presented in an intelligible fashion and written in standard English?

Reviewer #1: Yes

Reviewer #2: Yes

Reviewer #3: Yes

6. Review Comments to the Author

Reviewer #1: The authors have have submitted a revised version of their manuscript entitled “High-Resolution Gridded Estimates of Population Sociodemographics from the 2020 Census in California”. In their rebuttal letter, the authors provide extensive replies to all reviewer comments. In my view, all reviewer comments have been appropriately addressed with changes in the paper, rendering the manuscript acceptable for publication.

Specifically, key sections explaining the methods received clarifications; relevant citations were added; and an additional analysis to assess the results accuracy was conducted. These amendments help the reader to understand the presented dasymetric method and to assess the usefulness of the provided data set, CA-POP.

Reviewer #2: The manuscript has gone through important adaptations, demonstrating more clarity and coherence in the manner of presenting the results. I think this manuscript will be acceptable.

Reviewer #3: The author has addressed all my concerns, and I don’t have further questions. I recommend the publication of this manuscript with minor revisions. The author should carefully proofread the manuscript to avoid any mistakes or typos. Some potential grammatical errors and typos are listed below:

Line 24: Please unify the grammatical tenses (“showed” and “offers”)

Line 300: This sentence is not complete for ‘”population density limit” depiction.

Line 389: “Fresno, CA area” should be “Fresno area, CA”.

Line 424: “Satellite base imagery” should be “satellite-based imagery” or “satellite imagery”.

Line 490 and 510: Replace “table 2” by “table 3”.

Line 622: “…they are likely more even more accurate…”. Please delete the first “more”.

Line 623: Please add the proper preposition before “these final grids”.

7. PLOS authors have the option to publish the peer review history of their article (what does this mean?). If published, this will include your full peer review and any attached files.

Reviewer #1: No

Reviewer #2: No

Reviewer #3: No

---

## [Editor Report · Acceptance letter]

21 Jun 2022

PONE-D-22-01737R1 

High-resolution gridded estimates of population sociodemographics from the 2020 census in California 

Dear Dr. Depsky:

I'm pleased to inform you that your manuscript has been deemed suitable for publication in PLOS ONE. Congratulations! Your manuscript is now with our production department. 

Kind regards, 

on behalf of

Dr Krishna Prasad Vadrevu 

Academic Editor

PLOS ONE